# Penalising the biases in norm regularisation enforces sparsity

**Etienne Boursier**
INRIA CELESTE, LMO, Orsay, France
`etienne.boursier@inria.fr`

**Nicolas Flammarion**
TML Lab, EPFL, Switzerland
`nicolas.flammarion@epfl.ch`

## Abstract

Controlling the parameters' norm often yields good generalisation when training neural networks. Beyond simple intuitions, the relation between regularising parameters' norm and obtained estimators remains theoretically misunderstood. For one hidden ReLU layer networks with unidimensional data, this work shows the parameters' norm required to represent a function is given by the total variation of its second derivative, weighted by a $\sqrt{1 + x^2}$ factor. Notably, this weighting factor disappears when the norm of bias terms is not regularised. The presence of this additional weighting factor is of utmost significance as it is shown to enforce the uniqueness and sparsity (in the number of kinks) of the minimal norm interpolator. Conversely, omitting the bias' norm allows for non-sparse solutions. Penalising the bias terms in the regularisation, either explicitly or implicitly, thus leads to sparse estimators.

## 1 Introduction

Although modern neural networks are not particularly limited in terms of their number of parameters, they still demonstrate remarkable generalisation capabilities when applied to real-world data [Belkin et al., 2019, Zhang et al., 2021]. Intriguingly, both theoretical and empirical studies have indicated that the crucial factor determining the network's generalisation properties is not the sheer number of parameters, but rather the norm of these parameters [Bartlett, 1996, Neyshabur et al., 2014]. This norm is typically controlled through a combination of explicit regularisation techniques, such as weight decay [Krogh and Hertz, 1991], and some form of implicit regularisation resulting from the training algorithm employed [Soudry et al., 2018, Lyu and Li, 2019, Ji and Telgarsky, 2019, Chizat and Bach, 2020].

Neural networks with a large number of parameters can approximate any continuous function on a compact set [Barron, 1993]. Thus, without norm control, the space of estimated functions encompasses all continuous functions. In the parameter space, this implies considering neural networks with infinite width and unbounded weights [Neyshabur et al., 2014]. Yet, when weight control is enforced, the exact correspondence between the parameter space (i.e., the parameters $\theta$ of the network) and the function space (i.e., the estimated function $f_\theta$ produced by the network's output) becomes unclear. Establishing this correspondence is pivotal for comprehending the generalisation properties of overparameterised neural networks. Two fundamental questions arise.

**Question 1.** *What quantity in the function space, does the parameters' norm of a neural network correspond to?*

**Question 2.** *What functions are learnt when fitting training data with minimal parameters' norm?*

We study these questions in the context of a one-hidden ReLU layer network with a skip connection. Previous research [Kurková and Sanguineti, 2001, Bach, 2017] has examined generalisation guarantees for small representational cost functions, where the representational cost refers to the

norm required to parameterise the function. However, it remains challenging to interpret this representational cost using classical analysis tools and identify the corresponding function space. To address this issue, Question 1 seeks to determine whether this representational cost can be translated into a more interpretable functional (pseudo) norm. Note that Question 1 studies the parameters' norm required to fit a function on an entire domain. In contrast, when training a neural network for a regression task, we only fit a finite number of points given by the training data. Question 2 arises to investigate the properties of the learned functions when minimising some empirical loss with a regularisation of the parameters' norm regardless of whether it is done explicitly or implicitly.

In relation to our work, Savarese et al. [2019], Ongie et al. [2019] address Question 1 for one-hidden layer ReLU neural networks, focusing on univariate and multivariate functions, respectively. For a comprehensive review of this line of work, we recommend consulting the survey of Parhi and Nowak [2023]. On the other hand, Parhi and Nowak [2021], Debarre et al. [2022], Stewart et al. [2022] investigate Question 2 specifically in the univariate case. Additionally, Sanford et al. [2022] examine a particular multidimensional case. However, all of these existing studies overlook the bias parameters of the neural network when considering the $\ell_2$ regularisation term. By omitting the biases, the analysis and solutions to these questions become simpler.

In sharp contrast, our work addresses both Questions 1 and 2 for univariate functions *while also incorporating regularisation of the bias parameters*. It may appear as a minor detail—it is commonly believed that similar estimators are obtained whether or not the biases' norm[1] is penalised [see e.g. Ng, 2011]. Nonetheless, our research demonstrates that penalising the bias terms enforce sparsity and uniqueness of the estimated function, which is not achieved without including the bias regularisation. The practical similarity between these two explicit regularisations can be attributed to the presence of implicit regularisation, which considers the bias terms as well. The updates performed by first-order optimisation methods do not distinguish between bias and weight parameters, suggesting that they are subject to the same implicit regularisation. Consequently, while both regularisation approaches may yield similar estimators in practical settings, we contend that the theoretical estimators obtained with bias term regularisation capture the observed implicit regularisation effect. Hence, it is essential to investigate the implications of penalising the bias terms when addressing Questions 1 and 2, as the answers obtained in this scenario significantly differ from those without bias penalisation.

It is also worth mentioning that Shevchenko et al. [2022], Safran et al. [2022] prove that gradient flow learns sparse estimators for networks with ReLU activations. These sparsity guarantees are yet much weaker (larger number of activated directions) as they additionally deal with optimisation considerations (in opposition to directly considering the minimiser of the optimisation problem in both our work and the line of works mentioned above).

**Contributions.** After introducing the setting in Section 2, we address Question 1 in Section 3 using a similar analysis approach as Savarese et al. [2019]. The key result, Theorem 1, establishes that the representational cost of a function, when allowed a *free* skip connection, is given by the weighted total variation of its second derivative, incorporating a $\sqrt{1+x^2}$ term. Notably, penalising the bias terms introduces a $\sqrt{1+x^2}$ multiplicative weight in the total variation, contrasting with the absence of bias penalisation.

This weighting fundamentally impacts the answer to Question 2. In particular, it breaks the shift invariance property of the function's representational cost, rendering the analysis technique proposed by Debarre et al. [2022] inadequate[2]. To address this issue, we delve in Sections 4 and 5 into the computation and properties of solutions to the optimisation problem:

$$\inf_f \left\| \sqrt{1+x^2} f'' \right\|_{\mathrm{TV}} \quad \text{subject to } \forall i \in [n], \ f(x_i) = y_i.$$

In Section 4, we reformulate this problem as a continuous dynamic program, enabling a simpler analysis of the minimisation problem. Leveraging this dynamic program reformulation, Section 5 establishes the uniqueness of the solution. Additionally, under certain data assumptions, we demonstrate that the minimiser is among the sparsest interpolators in terms of the number of kinks. It is

---

[1]Even though Goodfellow et al. [2016, Chapter 7] claim that penalising the biases might lead to underfitting, our work does not focus on the optimisation aspect and assumes interpolation occurs.

[2]Although shift invariance is useful for analytical purposes, it is not necessarily desirable in practice. Notably, Nacson et al. [2022] show that dealing with uncentered data might be beneficial for learning relevant features, relying on the lack of shift invariance.

worth noting that similar results have been studied in the context of sparse spikes deconvolution [Candès and Fernandez-Granda, 2014, Fernandez-Granda, 2016, Poon et al., 2019], and our problem can be seen as a generalisation of basis pursuit [Chen et al., 2001] to infinite-dimensional parameter spaces. However, classical techniques for sparse spikes deconvolution are ill-suited for addressing Question 2, as the set of sparsest interpolators is infinite in our setting.

Finally, the significance of bias term regularisation in achieving sparser estimators during neural network training is illustrated on toy examples in Section 6. To ensure conciseness, only proof sketches are presented in the main paper, while the complete proofs can be found in the Appendix.

## 2   Infinite width networks

This section introduces the considered setting, representing unidimensional functions as infinite width networks. Some precise mathematical arguments are omitted here, since this construction follows directly the lines of Savarese et al. [2019], Ongie et al. [2019]. This work considers unidimensional functions $f_\theta : \mathbb{R} \to \mathbb{R}$ parameterised by a one hidden layer neural networks with ReLU activation as

$$f_\theta(x) = \sum_{j=1}^m a_j \sigma(w_j x + b_j),$$

where $\sigma(z) = \max(0, z)$ is the ReLU activation and $\theta = (a_j, w_j, b_j)_{j \in [m]} \in \mathbb{R}^{3m}$ are the parameters defining the neural network. The vector $\mathbf{a} = (a_j)_{j \in [m]}$ stands for the weights of the last layer, while $\mathbf{w}$ and $\mathbf{b}$ respectively stand for the weights and biases of the hidden layer. For any width $m$ and parameters $\theta$, the quantity of importance is the squared Euclidean norm of the parameters: $\|\theta\|_2^2 = \sum_{j=1}^m a_j^2 + w_j^2 + b_j^2$.

We recall that contrary to Savarese et al. [2019], Ongie et al. [2019], the bias terms are included in the considered norm here. We now define the representational cost of a function $f : \mathbb{R} \to \mathbb{R}$ as

$$R(f) = \inf_{\substack{m \in \mathbb{N} \\ \theta \in \mathbb{R}^{3m}}} \frac{1}{2} \|\theta\|_2^2 \quad \text{such that} \quad f_\theta = f.$$

By homogeneity of the parameterisation, a typical rescaling trick [see e.g. Neyshabur et al., 2014, Theorem 1] allows to rewrite

$$R(f) = \inf_{m, \theta \in \mathbb{R}^{3m}} \|\mathbf{a}\|_1 \quad \text{such that} \quad f_\theta = f \text{ and } w_j^2 + b_j^2 = 1 \text{ for any } j \in [m].$$

Note that $R(f)$ is only finite when the function $f$ is exactly described as a finite width neural network. We aim at extending this definition to a much larger functional space, i.e. to any function that can be arbitrarily well approximated by finite width networks, while keeping a (uniformly) bounded norm of the parameters. Despite approximating the function with finite width networks, the width necessarily grows to infinity when the approximation error goes to 0. Similarly to Ongie et al. [2019], define

$$\overline{R}(f) = \lim_{\varepsilon \to 0^+} \left( \inf_{m, \theta \in \mathbb{R}^{3m}} \frac{1}{2} \|\theta\|_2^2 \quad \text{such that} \quad |f_\theta(x) - f(x)| \leq \varepsilon \text{ for any } x \in [-1/\varepsilon, 1/\varepsilon] \right).$$

Note that the approximation has to be restricted to the compact set $[-1/\varepsilon, 1/\varepsilon]$ to avoid problematic degenerate situations. The functional space for which $\overline{R}(f)$ is finite is much larger than for $R$, and includes every compactly supported Lipschitz function, while coinciding with $R$ when the latter is finite.

By rescaling argument again, we can assume the hidden layer parameters $(w_j, b_j)$ are in $\mathbb{S}_1$ and instead consider the $\ell_1$ norm of the output layer weights. The parameters of a network can then be seen as a discrete signed measure on the unit sphere $\mathbb{S}_1$. When the width goes to infinity, a limit is then properly defined and corresponds to a possibly continuous signed measure. Mathematically, define $\mathcal{M}(\mathbb{S}_1)$ the space of signed measures $\mu$ on $\mathbb{S}_1$ with finite total variation $\|\mu\|_{\mathrm{TV}}$. Following the typical construction of Bengio et al. [2005], Bach [2017], an infinite width network is parameterised by a measure $\mu \in \mathcal{M}(\mathbb{S}_1)$ as[3]

$$f_\mu : x \mapsto \int_{\mathbb{S}_1} \sigma(wx + b) \mathrm{d}\mu(w, b). \tag{1}$$

Similarly to Ongie et al. [2019], $\overline{R}(f)$ verifies the equality

$$\overline{R}(f) = \inf_{\mu \in \mathcal{M}(\mathbb{S}_1)} \|\mu\|_{\mathrm{TV}} \text{ such that } f = f_\mu.$$

---

[3]By abuse of notation, we write both $f_\theta$ and $f_\mu$, as it is clear from context whether the subscript is a vector or a measure.

The right term defines the $\mathcal{F}_1$ norm [Kurková and Sanguineti, 2001], i.e. $\overline{R}(f) = \|f\|_{\mathcal{F}_1}$. The $\mathcal{F}_1$ norm is intuited to be of major significance for the empirical success of neural networks. In particular, generalisation properties of small $\mathcal{F}_1$ norm estimators are derived by Kurková and Sanguineti [2001], Bach [2017], while many theoretical results support that training one hidden layer neural networks with gradient descent often yields an implicit regularisation on the $\mathcal{F}_1$ norm of the estimator [Lyu and Li, 2019, Ji and Telgarsky, 2019, Chizat and Bach, 2020, Boursier et al., 2022]. However, this implicit regularisation of the $\mathcal{F}_1$ norm is not systematic and some works support that a different quantity can be implicitly regularised on specific examples [Razin and Cohen, 2020, Vardi and Shamir, 2021, Chistikov et al., 2023]. Still, $\mathcal{F}_1$ norm seems to be closely connected to the implicit bias and its significance is the main motivation of this paper. While previous works also studied the representational costs of functions by neural networks [Savarese et al., 2019, Ongie et al., 2019], they did not penalise the bias term in the parameters' norm, studying a functional norm slightly differing from the $\mathcal{F}_1$ norm. This subtlety is at the origin of different levels of sparsity between the obtained estimators with or without penalising the bias terms, as discussed in Sections 5 and 6; where sparsity of an estimator here refers to the minimal width required for a network to represent the function (or similarly to the cardinality of the support of $\mu$ in Equation (1)). This notion of sparsity is more meaningful than the sparsity of the parameters $\theta$ here, since different $\theta$ (with different levels of sparsity) can represent the exact same estimated function.

## 2.1 Unpenalised skip connection

Our objective is now to characterise the $\mathcal{F}_1$ norm of unidimensional functions and minimal norm interpolators, which can be approximately obtained when training a neural network with norm regularisation. The analysis and result yet remain complex despite the unidimensional setting. Allowing for an unpenalised affine term in the neural network representation leads to a cleaner characterisation of the norm and description of minimal norm interpolators. As a consequence, we parameterise in the remaining of this work finite and infinite width networks as follows:

$$f_{\theta,a_0,b_0} : x \mapsto a_0 x + b_0 + f_\theta(x), \quad \text{and} \quad f_{\mu,a_0,b_0} : x \mapsto a_0 x + b_0 + f_\mu(x),$$

where $(a_0, b_0) \in \mathbb{R}^2$. The affine part $a_0 x + b_0$ actually corresponds to a *free* skip connection in the neural network architecture [He et al., 2016] and allows to ignore the affine part in the representational cost of the function $f$, which we now define as

$$\overline{R}_1(f) = \lim_{\varepsilon \to 0^+} \left( \inf_{\substack{m,\theta \in \mathbb{R}^{3m} \\ (a_0,b_0) \in \mathbb{R}^2}} \frac{1}{2} \|\theta\|_2^2 \quad \text{such that} \quad |f_{\theta,a_0,b_0}(x) - f(x)| \leq \varepsilon \text{ for any } x \in [-\!1/\varepsilon, 1/\varepsilon] \right).$$

The representational cost $\overline{R}_1(f)$ is similar to $\overline{R}(f)$, but allows for a *free* affine term in the network architecture. Similarly to $\overline{R}(f)$, it can be proven, following the lines of Savarese et al. [2019], Ongie et al. [2019], that $\overline{R}_1(f)$ verifies

$$\overline{R}_1(f) = \inf_{\substack{\mu \in \mathcal{M}(\mathbb{S}_1) \\ a_0,b_0 \in \mathbb{R}}} \|\mu\|_{\mathrm{TV}} \quad \text{such that} \quad f = f_{\mu,a_0,b_0}.$$

The remaining of this work studies more closely the cost $\overline{R}_1(f)$. Theorem 1 in Section 3 can be directly extended to the cost $\overline{R}(f)$, i.e. without unpenalised skip connection. Its adapted version is given by Theorem 4 in Appendix C for completeness.

Multiple works also consider free skip connections as it allows for a simpler analysis [e.g. Savarese et al., 2019, Ongie et al., 2019, Debarre et al., 2022, Sanford et al., 2022]. Since a skip connection can be represented by two ReLU neurons ($z = \sigma(z) - \sigma(-z)$), it is commonly believed that considering a free skip connection does not alter the nature of the obtained results. This belief is further supported by empirical evidence in Section 6 and Appendix B, where our findings hold true both with and without free skip connections.

## 3 Representational cost

Theorem 1 below characterises the representational cost $\overline{R}_1(f)$ of any univariate function.

**Theorem 1.** *For any Lipschitz function* $f : \mathbb{R} \to \mathbb{R}$,

$$\overline{R}_1(f) = \left\| \sqrt{1 + x^2} f'' \right\|_{\mathrm{TV}} = \int_{\mathbb{R}} \sqrt{1 + x^2} \, \mathrm{d}|f''|(x).$$

*For any non-Lipschitz function, $\overline{R}_1(f) = \infty$.*

In Theorem 1, $f''$ is the distributional second derivative of $f$, which is well defined for Lipschitz functions. Without penalisation of the bias terms, the representational cost is given by the total variation of $f''$ [Savarese et al., 2019]. Theorem 1 states that penalising the biases adds a weight $\sqrt{1 + x^2}$ to $f''$. This weighting favors sparser estimators when training neural networks, as shown in Section 5. Also, the space of functions that can be represented by infinite width neural networks with finite parameters' norm, when the bias terms are ignored, corresponds to functions with bounded total variation of their second derivative. When including these bias terms in the representational cost, second derivatives additionally require a *light tail*. Without a *free* affine term, Theorem 4 in Appendix C characterises $\overline{R}(f)$, which yields an additional term accounting for the affine part of $f$.

We note that Remark 4.2 of E and Wojtowytsch [2021] and Theorem 1 by Li et al. [2020b] are closely related to Theorems 1 and 4. However, these results only establish an equivalence between the norm $\overline{R}(f)$ and another norm that quantifies the total variation of $\sqrt{1 + x^2} f''$, aside from the affine term. Our result, on the other hand, provides an exact equality between both norms, which proves to be particularly useful in the analysis of minimal norm interpolators.

**Example 1.** *If the function $f$ is given by a finite width network $f(x) = \sum_{i=1}^n a_i \sigma(w_i x + b_i)$ with $a_i, w_i \neq 0$ and pairwise different $\frac{b_i}{w_i}$; Theorem 1 yields $\overline{R}_1(f) = \sum_{i=1}^n |a_i| \sqrt{w_i^2 + b_i^2}$. This exactly corresponds to half of the squared $\ell_2$ norm of the vector $(c_i a_i, \frac{w_i}{c_i}, \frac{b_i}{c_i})_{i=1,\ldots,n}$ when $c_i = \sqrt{\frac{\sqrt{w_i + b_i^2}}{|a_i|}}$. This vector is thus a minimal representation of the function $f$.*

According to Theorem 1, the minimisation problem considered when training one hidden ReLU layer infinite width neural network with $\ell_2$ regularisation is equivalent to the minimisation problem

$$\inf_f \sum_{i=1}^n (f(x_i) - y_i)^2 + \lambda \left\| \sqrt{1 + x^2} f'' \right\|_{\mathrm{TV}}. \tag{2}$$

*What types of functions do minimise this problem? Which solutions does the $\left\| \sqrt{1 + x^2} f'' \right\|_{\mathrm{TV}}$ regularisation term favor?* These fundamental questions are studied in the following sections. We show that this regularisation favors functions that can be represented by small (finite) width neural networks. On the contrary, when the weight decay term does not penalise the biases of the neural network, such a sparsity is not particularly preferred as highlighted by Section 6.

## 4    Computing minimal norm interpolator

To study the properties of solutions obtained by training data with either an implicit or explicit weight decay regularisation, we consider the minimal norm interpolator problem

$$\inf_{\theta, a_0, b_0} \frac{1}{2} \|\theta\|_2^2 \quad \text{such that } \forall i \in [n], \ f_{\theta, a_0, b_0}(x_i) = y_i, \tag{3}$$

where $(x_i, y_i)_{i \in [n]} \in \mathbb{R}^{2n}$ is a training set. Without loss of generality, we assume in the following that the observations $x_i$ are ordered, i.e., $x_1 < x_2 < \ldots < x_n$. Thanks to Theorem 1, this problem is equivalent, when allowing infinite width networks, to

$$\inf_f \left\| \sqrt{1 + x^2} f'' \right\|_{\mathrm{TV}} \quad \text{such that } \forall i \in [n], \ f(x_i) = y_i. \tag{4}$$

Lemma 1 below actually makes these problems equivalent as soon as the width is larger than some threshold smaller than $n - 1$. Equation (4) then corresponds to Equation (2) when the regularisation parameter $\lambda$ is infinitely small.

**Lemma 1.** *The problem in Equation (4) admits a minimiser. Moreover, with $i_0 := \min\{i \in [n] | x_i \geq 0\}$, any minimiser is of the form*

$$f(x) = ax + b + \sum_{i=1}^{n-1} a_i (x - \tau_i)_+$$

*where $\tau_i \in (x_i, x_{i+1}]$ for any $i \in \{1, \ldots, i_0 - 2\}$, $\tau_{i_0-1} \in (x_{i_0-1}, x_{i_0})$ and $\tau_i \in [x_i, x_{i+1})$ for any $i \in \{i_0, \ldots, n-1\}$.*

Lemma 1 already provides a first guarantee on the sparsity of any minimiser of Equation (4). It indeed includes at most $n - 1$ kinks. In contrast, minimal norm interpolators with an infinite number

of kinks exist when the bias terms are not regularised [Debarre et al., 2022]. An even stronger sparse recovery result is given in Section 5. Lemma 1 can be seen as a particular case of Theorem 1 of Wang et al. [2021]. In the multivariate case and without a free skip connection, the latter states that the minimal norm interpolator has at most one kink (i.e. neuron) per *activation cone* of the weights[4] and has no more than $n + 1$ kinks in total. The idea of our proof is that several kinks among a single activation cone could be merged into a single kink in the same cone. The resulting function then still interpolates, but has a smaller representational cost.

Lemma 1 allows to only consider 2 parameters for each interval $(x_i, x_{i+1})$ (potentially closed at one end). Actually, the degree of freedom is only 1 on such intervals: choosing $a_i$ fixes $\tau_i$ (or inversely) because of the interpolation constraint. Lemma 2 below uses this idea to recast the minimisation Problem (4) as a dynamic program with unidimensional state variables $s_i \in \mathbb{R}$ for any $i \in [n]$.

**Lemma 2.** *If $x_1 < 0$ and $x_n \geq 0$, then we have for $i_0 = \min\{i \in [n] | x_i \geq 0\}$ the following equivalence of optimisation problems*

$$\min_{\substack{f \\ \forall i \in [n], f(x_i) = y_i}} \left\| \sqrt{1 + x^2} f'' \right\|_{\mathrm{TV}} = \min_{(s_{i_0-1}, s_{i_0}) \in \Lambda} g_{i_0}(s_{i_0}, s_{i_0-1}) + c_{i_0-1}(s_{i_0-1}) + c_{i_0}(s_{i_0}) \quad (5)$$

*where the set $\Lambda$ and the functions $g_i$ and $c_i$ are defined in Equations (6) to (8) below.*

Let us describe the dynamic program defining the functions $c_i$, which characterises the minimal norm interpolator thanks to Lemma 2. First define for any $i \in [n-1]$, the slope $\delta_i := \frac{y_{i+1} - y_i}{x_{i+1} - x_i}$; the function

$$g_{i+1}(s_{i+1}, s_i) := \sqrt{(x_{i+1}(s_{i+1} - \delta_i) - x_i(s_i - \delta_i))^2 + (s_{i+1} - s_i)^2} \text{ for any } (s_{i+1}, s_i) \in \mathbb{R}^2; \tag{6}$$

$$\text{and the intervals} \quad S_i(s) := \begin{cases} (-\infty, \delta_i] \text{ if } s > \delta_i \\ \{\delta_i\} \text{ if } s = \delta_i \\ [\delta_i, +\infty) \text{ if } s < \delta_i \end{cases} \quad \text{for any } s \in \mathbb{R}.$$

The set $\Lambda$ is then the union of three product spaces given by

$$\Lambda := (-\infty, \delta_{i_0-1}) \times (\delta_{i_0-1}, +\infty) \cup \{(\delta_{i_0-1}, \delta_{i_0-1})\} \cup (\delta_{i_0-1}, +\infty) \times (-\infty, \delta_{i_0-1}). \tag{7}$$

Finally, we define the functions $c_i : \mathbb{R} \to \mathbb{R}_+$ recursively as $c_1 = c_n \equiv 0$ and

$$\begin{aligned} c_{i+1} : s_{i+1} \mapsto \min_{s_i \in S_i(s_{i+1})} g_{i+1}(s_{i+1}, s_i) + c_i(s_i) \quad \text{for any } i \in \{1, \ldots, i_0 - 2\} \\ c_i : s_i \mapsto \min_{s_{i+1} \in S_i(s_i)} g_{i+1}(s_{i+1}, s_i) + c_{i+1}(s_{i+1}) \quad \text{for any } i \in \{i_0, \ldots, n-1\}. \end{aligned} \tag{8}$$

Equation (8) defines a dynamic program with a continuous state space. Intuitively for $i \geq i_0$, the variable $s_i$ accounts for the left derivative at the point $x_i$. The term $g_{i+1}(s_{i+1}, s_i)$ is the minimal cost (in neuron norm) for reaching the point $(x_{i+1}, y_{i+1})$ with a slope $s_{i+1}$, knowing that the left slope is $s_i$ at the point $(x_i, y_i)$. Similarly, the interval $S_i(s_i)$ gives the reachable slopes[5] at $x_{i+1}$, knowing the slope in $x_i$ is $s_i$. Finally, $c_i(s_i)$ holds for the minimal cost of fitting all the points $(x_{i+1}, y_{i+1}), \ldots, (x_n, y_n)$ when the left derivative in $(x_i, y_i)$ is given by $s_i$. It is defined recursively by minimising the sum of the cost for reaching the next point $(x_{i+1}, y_{i+1})$ with a slope $s_{i+1}$, given by $g_{i+1}(s_{i+1}, s_i)$; and the cost of fitting all the points after $x_{i+1}$, given by $c_{i+1}$. This recursive definition is illustrated in Figure 1 below. A symmetric definition holds for $i < i_0$.

The idea to derive Equation (6) is to first use Lemma 1 to get a finite representation of a minimal function $f^*$. From there, the minimal cost for connecting the point $(x_{i+1}, y_{i+1})$ with $(x_i, y_i)$ is done by using a single kink in between. The restrictions given by $s_i$ and $s_{i+1}$ then yield a unique possible kink. Minimizing its neuron norm then yields Equation (6)

**Remark 1.** *Equation (5) actually considers the junction of two dynamic programs: a first one corresponding to the points with negative $x$ values and a second one for positive values. This separation around $x = 0$ is not needed for Lemma 2, but allows for a cleaner analysis in Section 5. Lemmas 1 and 2 also hold for any arbitrary choice of $i_0$. In particular for $i_0 = 1$, Equation (5) would not consider the junction of two dynamic programs anymore, but a single one.*

---

[4]See Equation (21) in the Appendix for a mathematical definition.

[5]Here, a single kink is used in the interval $[x_i, x_{i+1}]$, thanks to Lemma 3.

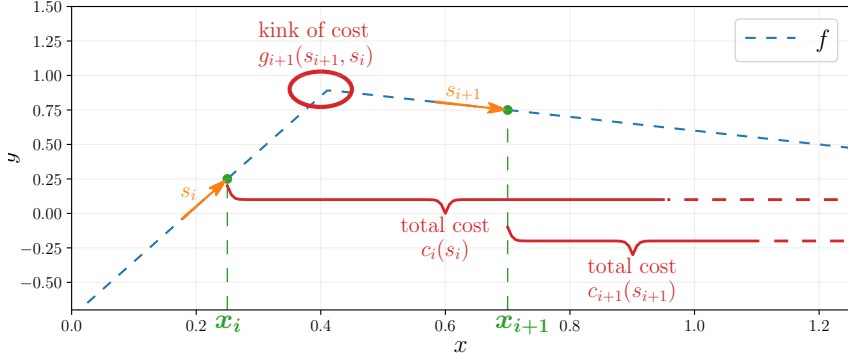

Figure 1: Recursive definition of the dynamic program for $i \geq i_0$.

**Remark 2.** *The assumption $x_1 < 0$ and $x_n \geq 0$ is not fundamental, but is only required to properly define the junction mentioned in Remark 1. If all the $x$ values are positive (or negative by symmetry), the analysis of the right term in Equation (5) is simplified, since there is no junction to consider. In particular, all the results from Section 5 hold without this assumption. These results are proven in the hardest case $x_1 < 0$ and $x_n \geq 0$ in Appendix E, from which other cases can be directly deferred.*

Lemma 2 formulates the minimisation of the representational cost among the interpolating functions as a simpler dynamic program on the sequence of slopes at each $x_i$. This equivalence is the key technical result of this work, from which Section 5 defers many properties on the minimiser(s) of Equation (4).

## 5 Properties of minimal norm interpolator

Thanks to the dynamic program formulation given by Lemma 2, this section derives key properties on the interpolating functions of minimal representational cost. In particular, it shows that Equation (4) always admits a unique minimum. Moreover, under some condition on the training set, this minimising function has the smallest number of kinks among the set of interpolators.

**Theorem 2.** *The following optimisation problem admits a unique minimiser:*

$$\inf_f \left\| \sqrt{1 + x^2} f'' \right\|_{\mathrm{TV}} \quad \text{such that } \forall i \in [n], \ f(x_i) = y_i.$$

The proof of Theorem 2 uses the correspondence between interpolating functions and sequences of slopes $(s_i)_{i \in [n]} \in \mathcal{S}$, where the set $\mathcal{S}$ is defined by Equation (23) in Appendix D.2. In particular, we show that the following problem admits a unique minimiser:

$$\min_{\mathbf{s} \in \mathcal{S}} \sum_{i=1}^{n-1} g_{i+1}(s_{i+1}, s_i). \tag{9}$$

We note in the following $\mathbf{s}^* \in \mathcal{S}$ the unique minimiser of the problem in Equation (9). From this sequence of slopes $\mathbf{s}^*$, the unique minimising function of Equation (4) can be recovered. Moreover, $\mathbf{s}^*$ minimises the dynamic program given by the functions $c_i$ as follows:

$$c_{i+1}(s_{i+1}^*) = g_{i+1}(s_{i+1}^*, s_i^*) + c_i(s_i^*) \quad \text{for any } i \in [i_0 - 2]$$
$$c_i(s_i^*) = g_{i+1}(s_{i+1}^*, s_i^*) + c_{i+1}(s_{i+1}^*) \quad \text{for any } i \in \{i_0, \ldots, n-1\}.$$

Using simple properties of the functions $c_i$ given by Lemma 7 in Appendix E, properties on $\mathbf{s}^*$ can be derived besides the uniqueness of the minimal norm interpolator. Lemma 3 below gives a first intuitive property of this minimiser, which proves helpful in showing the main result of the section.

**Lemma 3.** *For any $i \in [n]$, $s_i^* \in [\min(\delta_{i-1}, \delta_i), \max(\delta_{i-1}, \delta_i)]$, where $\delta_0 := \delta_1$ and $\delta_n := \delta_{n-1}$ by convention.*

A geometric interpretation of Lemma 3 is that the optimal (left or right) slope in $x_i$ is between the line joining $(x_{i-1}, y_{i-1})$ with $(x_i, y_i)$ and the line joining $(x_i, y_i)$ with $(x_{i+1}, y_{i+1})$.

### 5.1 Recovering a sparsest interpolator

We now aim at characterising when the minimiser of Equation (4) is among the set of sparsest interpolators, in terms of number of kinks. Before describing the minimal number of kinks required

to fit the data in Lemma 4, we partition $[x_1, x_n)$ into intervals of the form $[x_{n_k}, x_{n_{k+1}})$ where

$$n_0 = 1 \text{ and for any } k \geq 0 \text{ such that } n_k < n,$$

$$n_{k+1} = \min\left\{j \in \{n_k + 1, \ldots, n-1\} \mid \text{sign}(\delta_j - \delta_{j-1}) \neq \text{sign}(\delta_{j-1} - \delta_{j-2})\right\} \cup \{n\}, \quad (10)$$

and $\text{sign}(0) := 0$ by convention. If we note $f_{\text{lin}}$ the canonical piecewise linear interpolator, it is either convex, concave or affine on every interval $[x_{n_k-1}, x_{n_{k+1}}]$. This partitioning thus splits the space into convex, concave and affine parts of $f_{\text{lin}}$, as illustrated by Figure 2 on a toy example. This partition is crucial in describing the sparsest interpolators, thanks to Lemma 4.

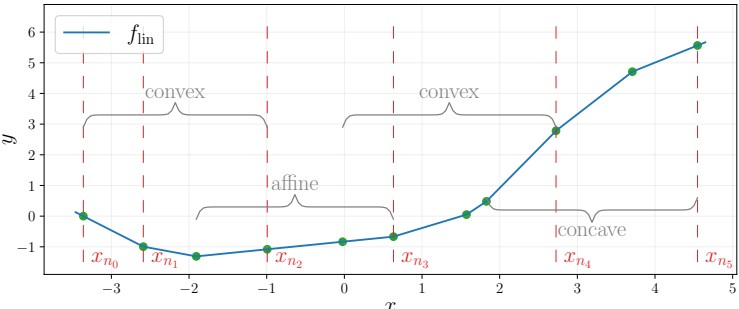

Figure 2: Partition given by $(n_k)_k$ on a toy example.

**Lemma 4.** *If we denote by $\|f''\|_0$ the cardinality of the support of the measure $f''$,*

$$\min_{\substack{f \\ \forall i, f(x_i) = y_i}} \|f''\|_0 = \sum_{k \geq 1} \left\lceil \frac{n_{k+1} - n_k}{2} \right\rceil \mathbb{1}_{\delta_{n_k-1} \neq \delta_{n_k}}.$$

Lemma 4's proof idea is that for any interval $[x_{k-1}, x_{k+1})$ where $f_{\text{lin}}$ is convex (resp. concave) non affine, any function requires at least one positive (resp. negative) kink to fit the three data points in this interval. The result then comes from counting the number of such disjoint intervals and showing that a specific interpolator exactly reaches this number.

The minimal number of kinks required to interpolate the data is given by Lemma 4. Before giving the main result of this section, we introduce the following assumption on the data $(x_k, y_k)_{k \in [n]}$.

**Assumption 1.** *For the sequence $(n_k)_k$ defined in Equation* (10)*:*

$$n_{k+1} - n_k \leq 3 \text{ or } \delta_{n_k} = \delta_{n_k - 1} \quad \text{for any } k \geq 0.$$

Assumption 1 exactly means there are no 6 (or more) consecutive points $x_k, \ldots, x_{k+5}$ such that $f_{\text{lin}}$ is convex (without 3 aligned points) or concave on $[x_k, x_{k+5}]$. This assumption depends a lot on the structure of the true model function (if there is any). For example, it holds if the truth is given by a piecewise linear function, while it may not if the truth is given by a quadratic function. Theorem 3 below shows that under Assumption 1, the minimal cost interpolator is amongst the sparsest interpolators, in number of its kinks.

**Theorem 3.** *If Assumption 1 holds, then*

$$\underset{\substack{f \\ \forall i, f(x_i) = y_i}}{\text{argmin}} \ \left\|\sqrt{1+x^2} f''\right\|_{\text{TV}} \in \ \underset{\substack{f \\ \forall i, f(x_i) = y_i}}{\text{argmin}} \ \|f''\|_0. \quad (11)$$

Theorem 3 states conditions under which the interpolating function $f$ with the smallest representational cost $\overline{R}_1(f)$ also has the minimal number of kinks, i.e. ReLU hidden neurons, among the set of interpolators. It illustrates how norm regularisation, and in particular adding the biases' norm to the weight decay, favors estimators with a small number of neurons. While training neural networks with norm regularisation, the final estimator can actually have many non-zero neurons, but they all align towards a few key directions. As a consequence, the obtained estimator is actually equivalent to a small width network, meaning they have the same output for every input $x \in \mathbb{R}$.

Recall that such a sparsity does not hold when the bias terms are not regularised. More precisely, some sparsest interpolators have a minimal representational cost in that case, but there are also minimal cost interpolators with an arbitrarily large (even infinite) number of kinks [Debarre et al., 2022]. There is thus no particular reason that the obtained estimator is sparse when minimising

the representational cost without penalising the bias terms. Section 6 empirically illustrates this difference of sparsity in the recovered estimators, depending on whether or not the bias parameters are penalised in the norm regularisation.

The generalisation benefit of this sparsity remains unclear. Indeed, generalisation bounds in the literature often rely on the parameters' norm rather than the network width (i.e., sparsity level). The relation between sparsest and min norm interpolators is important to understand in the particular context of implicit regularisation. In particular, while Boursier et al. [2022] conjectured that the implicit bias for regression problem was towards min norm interpolators, Chistikov et al. [2023] recently proved that the implicit bias could sometimes instead lead to sparsest interpolators. Our result suggests that both min norm and sparsest interpolators often coincide, which could explain the prior belief of convergence towards min norm interpolators. Yet, Theorem 3 and Chistikov et al. [2023] instead suggest that, at least in some situations, implicit bias favors sparsest interpolators, yielding different estimators[6].

**Remark 3.** *Theorem 3 states that sparse recovery, given by Equation* (11)*, occurs if Assumption 1 holds. When $n_{k+1} - n_k \geq 4$, i.e. there are convex regions of $f_{\mathrm{lin}}$ with at least 6 points, Appendix A gives a counterexample where Equation* (11) *does not hold. However, Equation* (11) *can still hold under weaker data assumptions than Assumption 1. In particular, Appendix A gives a necessary and sufficient condition for sparse recovery when there are convex regions with exactly 6 points. When we allow for convex regions with at least 7 points, it however becomes much harder to derive conditions where sparse recovery still occurs.*

**Remark 4.** *The counterexample presented in Appendix A reveals an unexpected outcome: minimal representational cost interpolators may not necessarily belong to the sparest interpolators. This finding is closely related to the idea that it may not be generally feasible to characterize the implicit regularisation of gradient descent as minimising parameters norm [Vardi and Shamir, 2021, Chistikov et al., 2023]. In particular, Vardi and Shamir [2021], Chistikov et al. [2023] rely on examples where minimal norm interpolators are not the sparsest ones; and the implicit regularisation instead favors the latter. We believe that this inherent limitation is one of the underlying reason for the different implicit regularization effects observed in other settings such as matrix factorization [Gunasekar et al., 2017, Razin and Cohen, 2020, Li et al., 2020a].*

## 5.2 Application to classification

In the binary classification setting, max-margin classifiers, defined as the minimiser of the problem

$$\min_f \overline{R}(f) \quad \text{such that } \forall i \in [n], y_i f(x_i) \geq 1, \tag{12}$$

are known to be the estimators of interest. Indeed, gradient descent on the cross entropy loss $l(\hat{y}, y) = \log(1 + e^{-\hat{y}y})$ converges in direction to such estimators [Lyu and Li, 2019, Chizat and Bach, 2020]. Theorem 3 can be used to characterise max- margin classifiers, leading to Corollary 1.

**Corollary 1.**

$$\underset{\substack{f \\ \forall i \in [n], y_i f(x_i) \geq 1}}{\operatorname{argmin}} \overline{R}_1(f) \subset \underset{\substack{f \\ \forall i \in [n], y_i f(x_i) \geq 1}}{\operatorname{argmin}} \|f''\|_0.$$

Theorem 3 yields that the max-margin classifier is among the sparsest margin classifiers, when a free skip connection is allowed. We believe that the left minimisation problem admits a unique solution. However, uniqueness cannot be directly derived from Theorem 3, but would instead require another thorough analysis, using an adapted dynamic programming reformulation. Since the uniqueness property is of minor interest, we here prefer to focus on a direct corollary of Theorem 3. We emphasise that no data assumptions are required for classification tasks, apart from being univariate.

# 6 Experiments

This section compares, through Figure 3, the estimators that are obtained with and without counting the bias terms in the regularisation, when training a one-hidden ReLU layer neural network. The code is made available at `github.com/eboursier/penalising_biases`.

---

[6]Note that this nuance only holds for regression tasks. Instead, it is known that implicit regularisation favors min norm interpolators in classification tasks [Chizat and Bach, 2020], which always coincide with sparsest interpolators in that case (see Corollary 1 in Section 5.2) for univariate data.

For this experiment, we train neural networks by minimising the empirical loss, regularised with the $\ell_2$ norm of the parameters (either with or without the bias terms) with a regularisation factor $\lambda = 10^{-3}$. Each neural network has $m = 200$ hidden neurons and all parameters are initialised i.i.d. as centered Gaussian variables of variance $1/\sqrt{m}$ (similar results are observed for larger initialisation scales).[7] There is no free skip connection here, which illustrates its benignity: the results that are expected by the above theory also happen without free skip connection. Experiments with a free skip connection are given in Appendix B and yield similar observations.

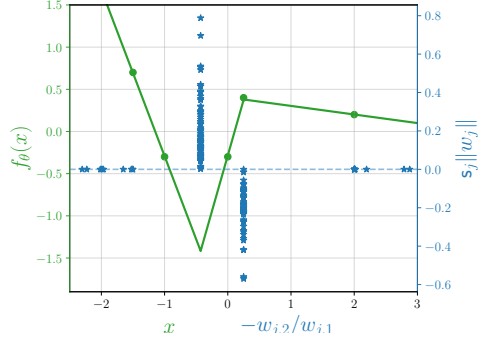

(a) Penalising bias terms in the $\ell_2$ regularisation.

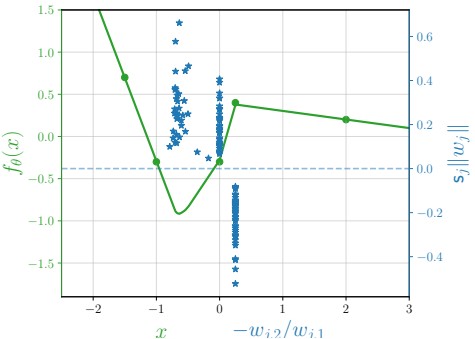

(b) Ignoring the bias terms in the $\ell_2$ regularisation.

Figure 3: Final estimator when training one-hidden layer network with $\ell_2$ regularisation. The green dots correspond to the data and the green line is the estimated function. Each blue star represents a hidden neuron $(w_j, b_j)$ of the network: its $x$-axis value is given by $-b_j/w_j$, which coincides with the position of the kink of its associated ReLU; its $y$-axis value is given by the output weight $a_j$.

As predicted by our theoretical study, penalising the bias terms in the $\ell_2$ regularisation enforces the sparsity of the final estimator. The estimator of Figure 3a indeed counts 2 kinks (the smallest number required to fit the data), while in Figure 3b, the directions of the neurons are scattered. More precisely, the estimator is almost *smooth* near $x = -0.5$, while the sparse estimator of Figure 3a is clearly not differentiable at this point. Also, the estimator of Figure 3b includes a clear additional kink at $x = 0$. Figure 3 thus illustrates that counting the bias terms in regularisation can lead to sparser estimators.

# 7 Conclusion

This work studies the importance of parameters' norm for one hidden ReLU layer neural networks in the univariate case. In particular, the parameters' norm required to represent a function is given by $\left\| \sqrt{1 + x^2} f'' \right\|_{\mathrm{TV}}$ when allowing for a free skip connection. In comparison to weight decay, which omits the bias parameters in the norm, an additional $\sqrt{1 + x^2}$ weighting term appears in the representational cost. This weighting is of crucial importance since it implies uniqueness of the minimal norm interpolator. Moreover, it favors sparsity of this interpolator in number of kinks. Minimising the parameters' norm (with the biases), which can be either obtained by explicit or implicit regularisation when training neural networks, thus leads to sparse interpolators. We believe this sparsity is a reason for the good generalisation properties of neural networks observed in practice.

Although these results provide some understanding of minimal norm interpolators, extending them to more general and difficult settings remains open. Even if the representational cost might be described in the multivariate case [as done by Ongie et al., 2019, without bias penalisation], characterising minimal norm interpolators seems very challenging in that case. Characterising minimal norm interpolators, with no free skip connection, also presents a major challenge for future work.

---

[7]For small initialisations, both methods yield sparse estimators, since implicit regularisation of the bias terms is significant in that case. Our goal is only to illustrate the differences in the minimisers of the two problems (with and without bias penalisation), without any optimisation consideration.

## Acknowledgments and Disclosure of Funding

The authors thank Gal Vardi, for suggesting the proof of Corollary 1, through a direct use of Theorem 3. The authors also thank Claire Boyer, Julien Fageot and Loucas Pillaud-Vivien for very helpful discussions and feedback.

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

## A Discussing Assumption 1

Theorem 3 requires Assumption 1, which assumes that there are no convex (or concave) regions of $f_{\text{lin}}$ with at least 6 data points. Actually, when there is a convex (or concave) region with exactly 6 data points, i.e. $n_{k+1} = n_k + 4$, Theorem 3 holds (for this region) if and only if for $i = n_k + 1$:

$$\frac{\langle u_i, w_{i-1}\rangle \langle u_{i+1}, w_{i+1}\rangle}{\|w_{i-1}\| \, \|w_{i+1}\|} - \langle u_i, u_{i+1}\rangle \leq \sqrt{\|u_i\|^2 - \frac{\langle u_i, w_{i-1}\rangle^2}{\|w_{i-1}\|^2}} \sqrt{\|u_{i+1}\|^2 - \frac{\langle u_{i+1}, w_{i+1}\rangle^2}{\|w_{i+1}\|^2}}$$
(13)

$$\text{where} \quad u_i = (x_i, 1); \quad w_{i-1} = \frac{\delta_i - \delta_{i-1}}{\delta_i - \delta_{i-2}}(x_i, 1) + \frac{\delta_{i-1} - \delta_{i-2}}{\delta_i - \delta_{i-2}}(x_{i-1}, 1);$$

$$\text{and} \quad u_{i+1} = (x_{i+1}, 1); \quad w_{i+1} = \frac{\delta_{i+2} - \delta_{i+1}}{\delta_{i+2} - \delta_i}(x_{i+2}, 1) + \frac{\delta_{i+1} - \delta_i}{\delta_{i+2} - \delta_i}(x_{i+1}, 1).$$

The proof of this result (omitted here) shows that the problem

$$\min_{(s_i, s_{i+1}) \in [\delta_{i-1}, \delta_i] \times [\delta_i, \delta_{i+1}]} g_i(s_i, s^*_{i-1}) + g_{i+1}(s_{i+1,s_i}) + g_{i+2}(s^*_{i+2}, s_{i+1})$$

is minimised for $(s_i, s_{i+1}) = (\delta_i, \delta_i)$ if and only if Equation (13) holds, which corresponds to the (unique) sparsest way to interpolate the data on this convex region. To show that the minimum is reached at that point, we can first notice that $s^*_{i-1} = \delta_{i-2}$ and $s^*_{i+2} = \delta_{i+2}$. Then, it requires a meticulous study of the directional derivatives of the (convex but non-differentiable) objective function at the point $(\delta_i, \delta_i)$.

Figure 4 below illustrates a case of 6 data points, where the condition of Equation (13) does not hold. Clearly, the minimal norm interpolator differs from the (unique) sparsest interpolator in that case.

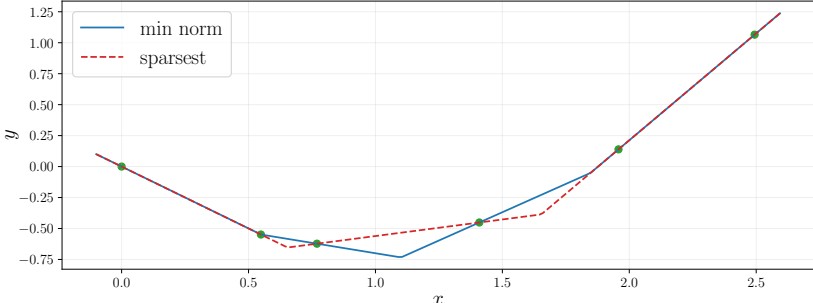

Figure 4: Case of difference between minimal norm interpolator and sparsest interpolator.

When considering more than 6 points, studying the minimisation problem becomes cumbersome and no simple condition of sparse recovery can be derived. When generating random data with large convex regions, e.g. 35 points, the minimal norm interpolator is rarely among the sparsest interpolators. Moreover, it seems that its number of kinks could be arbitrarily close to 34, which is the trivial upper bound of the number of kinks given by Lemma 3; while the sparsest interpolators only have 17 kinks.

## B Additional experiments

Figure 5 shows the minimiser of Equation (4) on the toy example of Figure 2. The minimising function is computed thanks to the dynamic program given by Lemma 2. Although the variables of this dynamic program are continuous, we can efficiently solve it by approximating the constraint space of each slope $s_i$ as a discrete grid of $[\delta_{i-1}, \delta_i]$ thanks to Lemma 3. For the data used in Figure 5, Assumption 1 holds. It is clear that the minimiser is very sparse, counting only 4 kinks. The partition given by Figure 2 then shows that this is indeed the smaller possible number of kinks, thanks to Lemma 4. On the other hand, the canonical piecewise linear interpolator $f_{\text{lin}}$ is is not as sparse and counts 7 kinks here.

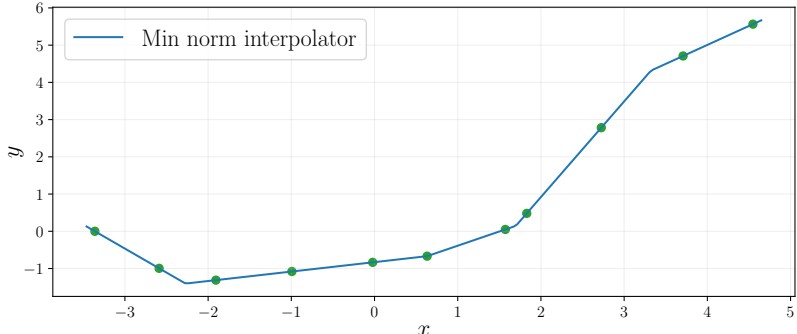

Figure 5: Minimiser of Equation (4) on a toy data example.

Figure 6 considers the exact same setting as Figure 3 in Section 6. The only difference is that we here allow a free skip connection in the neural network architecture, which represents the setting exactly described by Theorem 3.

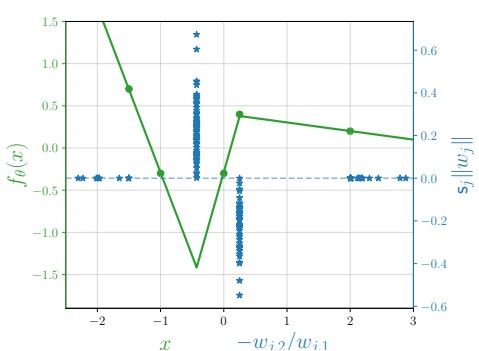

(a) Final estimator when penalising bias terms in the $\ell_2$ regularisation.

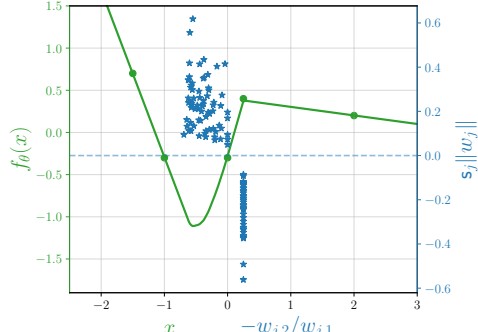

(b) Final estimators when training one-hidden ReLU neural networks with $\ell_2$ regularisation **and a free skip connection**.

Figure 6: Final estimators when training one-hidden ReLU neural networks with $\ell_2$ regularisation. The green dots correspond to the data, while the green line is the estimated function. Each blue star represents a hidden neuron $(w_j, b_j)$ of the network: its $x$-axis value is given by $-b_j/w_j$, which coincides with the position of the kink of its associated ReLU; its $y$-axis value is given by the output layer weight $a_j$.

Similar observations can be made: the obtained estimator when counting the bias terms in regularisation only has 2 kinks, while the estimator obtained by omitting the biases in the regularisation is much smoother (and thus much less sparse in the number of kinks). The only difference is that the latter estimator here does not have a clear kink at $x = 0$, but is instead even smoother on the interval $[-0.5, 0]$. This is explained by the presence of more scattered kinks in this interval. Despite this slight difference, the main observation remains unchanged: the estimator is a sparsest one when counting the bias terms, while it counts a lot of kinks (and is even smooth) when omitting the biases.

## C   Proofs of Section 3

Theorem 4 below extends the characterisation of the representational cost $\overline{R}_1(f)$ of Theorem 1.

**Theorem 4.** *For any Lipschitz function $f : \mathbb{R} \to \mathbb{R}$,*

$$\overline{R}_1(f) = \left\| \sqrt{1 + x^2} f'' \right\|_{\mathrm{TV}} = \int_{\mathbb{R}} \sqrt{1 + x^2} \, \mathrm{d}|f''|(x)$$

$$\text{and} \quad \overline{R}(f) = \left\| \sqrt{1 + x^2} f'' \right\|_{\mathrm{TV}} + D(x_f, \mathcal{C}_f),$$

*where*

$$x_f = \left( f'(+\infty) + f'(-\infty), f(0) - \int_{\mathbb{R}} |x| \mathrm{d}f''(x) \right)$$

$$\mathcal{C}_f = \left\{ \int_{\mathbb{R}} \varphi(x) \mathrm{d}f''(x), -\int_{\mathbb{R}} x\varphi(x) \mathrm{d}f''(x) \mid \|\varphi\|_\infty \le 1 \right\}$$

$$D(x_f, \mathcal{C}_f) = \inf_{x \in \mathcal{C}_f} \|x - x_f\|.$$

*For any non-Lipschitz function, $\overline{R}_1(f) = \overline{R}(f) = +\infty$.*

*Proof.* We only prove the equality on $\overline{R}(f)$ here. The other part of Theorem 4 can be directly deduced from this proof. First assume that $\overline{R}(f)$ is finite. We can then consider some $\mu \in \mathcal{M}(\mathbb{S}_1)$ such that for any $x \in \mathbb{R}$, $f(x) = \int_{\mathbb{S}_1} \sigma(wx + b) \mathrm{d}\mu(w, b)$. Note that $f$ is necessarily $\|\mu\|_{\mathrm{TV}}$-Lipschitz, which proves the second part of Theorem 4. Without loss of generality, we can parameterise $\mathbb{S}_1$ on $\theta \in [-\frac{\pi}{2}, \frac{3\pi}{2})$ with $T^{-1}(\theta) = (\cos\theta, \sin\theta)$. If we note $\nu = T_\# \mu$ the pushforward measure of $\mu$ by $T$, we then have

$$f(x) = \int_{-\frac{\pi}{2}}^{\frac{3\pi}{2}} \sigma(x\cos\theta + \sin\theta) \mathrm{d}\nu(\theta). \tag{14}$$

Since the total variation of $\mu$ and thus $\nu$ is bounded, we can derive under the integral sign:

$$f'(x) = \int_{-\frac{\pi}{2}}^{\frac{3\pi}{2}} \cos\theta \mathbb{1}_{x\cos\theta + \sin\theta \ge 0} \mathrm{d}\nu(\theta).$$

Now note that $x\cos\theta + \sin\theta \ge 0$ if and only if $\theta \in [-\arctan(x), \pi - \arctan(x)]$ since $\theta \in [-\frac{\pi}{2}, \frac{3\pi}{2})$, i.e.

$$f'(x) = \int_{-\arctan x}^{\pi - \arctan x} \cos\theta \mathrm{d}\nu(\theta).$$

When deriving this expression over $x$, we finally get for the distribution $f''$

$$\mathrm{d}f''(x) = -\frac{\cos(\pi - \arctan(x))\, \mathrm{d}\nu(\pi - \arctan(x)) - \cos(-\arctan(x))\, \mathrm{d}\nu(-\arctan(x))}{1 + x^2}$$

$$= \frac{\cos(\arctan(x))(\mathrm{d}\nu(\pi - \arctan(x)) + \mathrm{d}\nu(-\arctan(x)))}{1 + x^2}.$$

This equality is straightforward for continuous distributions $\nu$. Extending it to any distribution $\nu$ requires some extra work but can be obtained following the typical definition of distributional derivative.

Defining $\nu_+(\theta) = \nu(\theta) + \nu(\pi + \theta)$ for any $\theta \in [-\frac{\pi}{2}, \frac{\pi}{2})$ and noting that $\cos(\arctan x) = \frac{1}{\sqrt{1+x^2}}$,

$$\forall x \in \mathbb{R}, \sqrt{1 + x^2} \mathrm{d}f''(x) = \frac{\mathrm{d}\nu_+(-\arctan(x))}{1 + x^2}.$$

Or equivalently, for any $\theta \in (-\frac{\pi}{2}, \frac{\pi}{2})$

$$\frac{\mathrm{d}f''(-\tan\theta)}{\cos\theta} = \cos^2(\theta)\, \mathrm{d}\nu_+(\theta). \tag{15}$$

Similarly to the proof of Savarese et al. [2019], $f''$ fixes $\nu_+$ and the only degree of freedom is on $\nu_\perp(\theta) := \nu(\theta) - \nu(\pi + \theta)$. The proof now determines which valid $\nu_\perp$ minimises $\|\mu\|_{\mathrm{TV}} = \|\nu\|_{\mathrm{TV}}$. Equation (14) implies the following condition on $\nu_\perp$

$$f(x) = \frac{1}{2} \int_{-\frac{\pi}{2}}^{\frac{\pi}{2}} \sigma(-x\cos\theta - \sin\theta) \mathrm{d}(\nu_+ + \nu_\perp)(\theta) + \frac{1}{2} \int_{-\frac{\pi}{2}}^{\frac{\pi}{2}} \sigma(x\cos\theta + \sin\theta) \mathrm{d}(\nu_+ - \nu_\perp)(\theta)$$

$$= \frac{1}{2} \int_{-\frac{\pi}{2}}^{\frac{\pi}{2}} |x\cos\theta + \sin\theta| \mathrm{d}\nu_+(\theta) + \frac{x}{2} \int_{-\frac{\pi}{2}}^{\frac{\pi}{2}} \cos\theta \mathrm{d}\nu_\perp(\theta) + \frac{1}{2} \int_{-\frac{\pi}{2}}^{\frac{\pi}{2}} \sin\theta \mathrm{d}\nu_\perp(\theta).$$

While $\nu_+$ is given by $f''$, $\nu_\perp$ holds for affine part of $f$. The above equality directly leads to the following condition on $\nu_\perp$

$$\begin{cases} \int_{-\frac{\pi}{2}}^{\frac{\pi}{2}} \cos\theta\, \mathrm{d}\nu_\perp(\theta) = f'(+\infty) + f'(\infty) \\ \int_{-\frac{\pi}{2}}^{\frac{\pi}{2}} \sin\theta\, \mathrm{d}\nu_\perp(\theta) = f(0) - \int_{-\frac{\pi}{2}}^{\frac{\pi}{2}} |\sin\theta| \mathrm{d}\nu_+(\theta) \end{cases} . \tag{16}$$

Now note that $2 \|\mu\|_{\mathrm{TV}} = \|\nu_+ + \nu_\perp\|_{\mathrm{TV}} + \|\nu_+ - \nu_\perp\|_{\mathrm{TV}}$, so that $\overline{R}(f)$ is given by

$$2\overline{R}(f) = \min_{\nu_\perp} \|\nu_+ + \nu_\perp\|_{\mathrm{TV}} + \|\nu_+ - \nu_\perp\|_{\mathrm{TV}} \quad \text{such that } \nu_\perp \text{ verifies Equation (16)}.$$

Lemma 5 below then implies[8]

$$\overline{R}(f) = \|\nu_+\|_{\mathrm{TV}} + D(x_f, \mathcal{C}_f),$$

where $x_f$ and $\mathcal{C}_f$ are defined in Theorem 4. Equation (15) leads with a simple change of variable when $\overline{R}(f)$ is finite to

$$\overline{R}(f) = \left\|\sqrt{1 + x^2} f''\right\|_{\mathrm{TV}} + D(x_f, \mathcal{C}_f).$$

Reciprocally, when $\left\|\sqrt{1 + x^2} f''\right\|_{\mathrm{TV}}$ is finite, we can define $\nu_+$ as in Equation (15) and $\nu_\perp$ as a sum of Diracs in $-\frac{\pi}{2}$ and $0$ verifying Equation (16). The corresponding $\mu$ is then of finite total variation, implying that $\overline{R}(f)$ is finite. This ends the proof of the first part of Theorem 4:

$$\overline{R}(f) = \left\|\sqrt{1 + x^2} f''\right\|_{\mathrm{TV}} + D(x_f, \mathcal{C}_f).$$

For $\overline{R}_1(f)$, the analysis is simpler since there is no constraint on $\nu_\perp$, whose optimal choice is then given by $\nu_\perp = 0$. $\qquad\square$

**Lemma 5.** *The minimisation program*

$$
\begin{aligned}
\min_{\nu_\perp} &\int_{-\frac{\pi}{2}}^{\frac{\pi}{2}} \mathrm{d}|\nu_+ + \nu_\perp| + \int_{-\frac{\pi}{2}}^{\frac{\pi}{2}} \mathrm{d}|\nu_+ - \nu_\perp| \\
\textit{such that } &\left(\int_{-\pi/2}^{\pi/2} \cos\theta \, \mathrm{d}\nu_\perp(\theta), \int_{-\pi/2}^{\pi/2} \sin\theta \, \mathrm{d}\nu_\perp(\theta)\right) = (a, b)
\end{aligned}
\tag{17}
$$

*is equivalent to*

$$2\int_{-\frac{\pi}{2}}^{\frac{\pi}{2}} \mathrm{d}|\nu_+| + 2\min_{u \in \mathcal{C}} \|(a, b) - u\|,$$

$$\textit{where } \mathcal{C} = \left\{ \left(\int_{-\frac{\pi}{2}}^{\frac{\pi}{2}} \cos(\theta)\varphi(\theta)\mathrm{d}\nu_+(\theta), \int_{-\frac{\pi}{2}}^{\frac{\pi}{2}} \sin(\theta)\varphi(\theta)\mathrm{d}\nu_+(\theta)\right) \mid \|\varphi\|_\infty \le 1 \right\}.
\tag{18}
$$

*Proof.* For any $\nu_\perp$ in the constraint set of Equation (17), we can use a decomposition $\nu_\perp = \varphi\nu_+ + \mu_2$ where $\|\varphi\|_\infty \le 1$. It then comes pointwise

$$|\nu_+ + \nu_\perp| + |\nu_+ - \nu_\perp| \le 2|\mu_2| + |(1 + \varphi)\nu_+| + |(1 - \varphi)\nu_+| \tag{19}$$
$$= 2|\mu_2| + 2|\nu_+|.$$

As a consequence, if we note $v$ the infimum given by Equation (17):

$$v \le 2\int_{-\frac{\pi}{2}}^{\frac{\pi}{2}} \mathrm{d}|\nu_+| + 2\min_{(\varphi, \mu_2) \in \Gamma} \int_{-\frac{\pi}{2}}^{\frac{\pi}{2}} \mathrm{d}|\mu_2|,$$

where

$$\Gamma = \left\{ (\varphi, \mu_2) \mid \|\varphi\|_\infty \le 1 \text{ and } \int_{-\pi/2}^{\pi/2} (\cos\theta, \sin\theta) \, (\varphi(\theta)\mathrm{d}\nu_+(\theta) + \mathrm{d}\mu_2(\theta)) = (a, b) \right\}.$$

Moreover for a fixed $\nu_\perp$, we can choose $(\varphi, \mu_2)$ as:

$$
\begin{cases}
\varphi = \mathrm{sign}(\frac{\mathrm{d}\nu_\perp}{\mathrm{d}\nu_+}) \min\left(\left|\frac{\mathrm{d}\nu_\perp}{\mathrm{d}\nu_+}\right|, 1\right), \\
\mu_2 = \nu_\perp - \varphi\nu_+,
\end{cases}
$$

where $\frac{\mathrm{d}\nu_\perp}{\mathrm{d}\nu_+}$ denotes by abuse of notation the Radon-Nikodym derivative $\frac{\mathrm{d}\nu_a}{\mathrm{d}\nu_+}$, with the Lebesgue decomposition $\nu_\perp = \nu_a + \nu_s$ with $\nu_a \ll \nu_+$ and $\nu_s \perp \nu_+$. For this choice, Equation (19) becomes an equality, which directly implies that

$$v = 2\int_{-\frac{\pi}{2}}^{\frac{\pi}{2}} \mathrm{d}|\nu_+| + 2\min_{(\varphi, \mu_2) \in \Gamma} \int_{-\frac{\pi}{2}}^{\frac{\pi}{2}} \mathrm{d}|\mu_2|.$$

---

[8]A change of variable is also necessary to observe that the $\min$ of Lemma 5 is equal to $D(x_f, \mathcal{C}_f)$.

It now remains to prove that $\min_{u \in \mathcal{C}} \|(a,b) - u\|^2 = \min_{(\varphi, \mu_2) \in \Gamma} \int_{-\frac{\pi}{2}}^{\frac{\pi}{2}} \mathrm{d}|\mu_2|$. Fix in the following $\varphi$ such that $\|\varphi\|_\infty \leq 1$ and note

$$\begin{cases} x = a - \int_{-\pi/2}^{\pi/2} \cos\theta\,\varphi(\theta)\mathrm{d}\nu_+(\theta), \\ y = b - \int_{-\pi/2}^{\pi/2} \sin\theta\,\varphi(\theta)\mathrm{d}\nu_+(\theta). \end{cases}$$

It now suffices to show that for any fixed $\varphi$:

$$\min_{\mu_2 \text{ s.t. } (\varphi,\mu_2) \in \Gamma} \int_{-\frac{\pi}{2}}^{\frac{\pi}{2}} \mathrm{d}|\mu_2| = \|(x,y)\|.$$

The constraint set is actually $\left\{ \mu_2 \mid \int_{-\pi/2}^{\pi/2} (\cos\theta, \sin\theta)\,\mathrm{d}\mu_2(\theta) = (x,y) \right\}$. Now define

$$\theta^* = \arcsin\left( \frac{\operatorname{sign}(x)y}{\sqrt{x^2+y^2}} \right) \quad \text{and} \quad \mu_2^* = \operatorname{sign}(x)\sqrt{x^2+y^2}\delta_{\theta^*},$$

where $\delta_{\theta^*}$ is the Dirac distribution located at $\theta^*$. This definition is only valid if $x \neq 0$, otherwise we choose $\mu_2^* = -y\delta_{-\frac{\pi}{2}}$.

Note that $\mu_2^*$ is in the constraint set and $\int_{-\frac{\pi}{2}}^{\frac{\pi}{2}} \mathrm{d}|\mu_2| = \|(x,y)\|$, i.e.

$$\min_{\mu_2 \text{ s.t. } (\varphi,\mu_2) \in \Gamma} \int_{-\frac{\pi}{2}}^{\frac{\pi}{2}} \mathrm{d}|\mu_2| \leq \|(x,y)\|.$$

Now consider any $\mu_2$ in the constraint set and decompose $\mu_2 = \mu_2^+ - \mu_2^-$ with $(\mu_2^+, \mu_2^-) \in \mathcal{M}_+([-\frac{\pi}{2}, \frac{\pi}{2}))^2$. Define

$$(x_+, y_+) = \left( \int_{-\frac{\pi}{2}}^{\frac{\pi}{2}} \cos\theta\,\mathrm{d}\mu_2^+, \int_{-\frac{\pi}{2}}^{\frac{\pi}{2}} \sin\theta\,\mathrm{d}\mu_2^+ \right)$$

$$(x_-, y_-) = \left( \int_{-\frac{\pi}{2}}^{\frac{\pi}{2}} \cos\theta\,\mathrm{d}\mu_2^-, \int_{-\frac{\pi}{2}}^{\frac{\pi}{2}} \sin\theta\,\mathrm{d}\mu_2^- \right)$$

By Cauchy-Schwarz inequality,

$$\int \cos^2(\theta)\,\mathrm{d}\mu_2^+(\theta) \int \mathrm{d}\mu_2^+(\theta) \geq x_+^2,$$

$$\int \sin^2(\theta)\,\mathrm{d}\mu_2^+(\theta) \int \mathrm{d}\mu_2^+(\theta) \geq y_+^2.$$

Summing these two inequalities yields

$$\int \mathrm{d}\mu_2^+ \geq \sqrt{x_+^2 + y_+^2}.$$

Similarly, we have

$$\int \mathrm{d}\mu_2^- \geq \sqrt{x_-^2 + y_-^2}.$$

Recall that $\int \mathrm{d}|\mu_2| = \int \mathrm{d}\mu_2^+ + \int \mathrm{d}\mu_2^-$. By triangle inequality, this yields:

$$\int \mathrm{d}|\mu_2| \geq \|(x_+, y_+)\| + \|(x_-, y_-)\|$$

$$\geq \|(x_+, y_+) - (x_-, y_-)\| = \|(x,y)\|.$$

As a consequence:

$$\min_{\mu_2 \text{ s.t. } (\varphi,\mu_2) \in \Gamma} \int_{-\frac{\pi}{2}}^{\frac{\pi}{2}} \mathrm{d}|\mu_2| \geq \|(x,y)\|.$$

We finally showed that

$$\min_{\mu_2 \text{ s.t. } (\varphi,\mu_2) \in \Gamma} \int_{-\frac{\pi}{2}}^{\frac{\pi}{2}} \mathrm{d}|\mu_2| = \left\| (a,b) - \left( \int_{-\pi/2}^{\pi/2} \cos\theta\,\varphi(\theta)\mathrm{d}\nu_+(\theta), \int_{-\pi/2}^{\pi/2} \sin\theta\,\varphi(\theta)\mathrm{d}\nu_+(\theta) \right) \right\|.$$

This leads to Lemma 5 when taking the infimum over $\varphi$. $\qquad\square$

## D   Proof of Section 4

### D.1   Proof of Lemma 1

We first need to show the existence of a minimum. Using the definition of $\overline{R}_1(f)$ and Theorem 1, Equation (4) is equivalent to

$$\inf_{\mu,a,b} \|\mu\|_{\mathrm{TV}} \quad \text{such that for any } i \in [n], \ f_{\mu,a,b}(x_i) = y_i. \tag{20}$$

Consider a sequence $(\mu_j, a_j, b_j)_j$ such that $f_{\mu_j,a_j,b_j}(x_i) = y_i$ for any $i$ and $j$ and $\|\mu_j\|_{\mathrm{TV}}$ converges to the infimum of Equation (20). The sequence $\|\mu_j\|_{\mathrm{TV}}$ is necessarily bounded. This also implies that both $(a_j)$ and $(b_j)$ are bounded[9]. Since the space of finite signed measures on $\mathbb{S}_1$ is a Banach space, there is a subsequence converging weakly towards some $(\mu, a, b)$. By weak convergence, $(\mu, a, b)$ is in the constraints set of Equation (20) and $\|\mu\|_{\mathrm{TV}} = \lim_j \|\mu_j\|_{\mathrm{TV}}$. $(\mu, a, b)$ is thus a minimiser of Equation (20). We thus proved the existence of a minimum for Equation (4), which is reached for $f_{\mu,a,b}$.

Define for the sake of the proof the activation cones $C_i$ as

$$
\begin{aligned}
C_0 &= \left\{ \theta \in \mathbb{R}^2 \mid \forall i = 1, \dots, n, \langle \theta, (x_i, 1) \rangle \geq 0 \right\}, \\
C_i &= \left\{ \theta \in \mathbb{R}^2 \mid \langle \theta, (x_{i+1}, 1) \rangle \geq 0 > \langle \theta, (x_i, 1) \rangle \right\} \quad \text{for any } i = 1, \dots, i_0 - 2, \\
C_{i_0-1} &= \left\{ \theta \in \mathbb{R}^2 \mid \langle \theta, (x_{i_0}, 1) \rangle > 0 > \langle \theta, (x_{i_0-1}, 1) \rangle \right\}, \\
C_i &= \left\{ \theta \in \mathbb{R}^2 \mid \langle \theta, (x_{i+1}, 1) \rangle > 0 \geq \langle \theta, (x_i, 1) \rangle \right\} \quad \text{for any } i = i_0, \dots, n - 1, \\
C_n &= \left\{ \theta \in \mathbb{R}^2 \setminus \{0\} \mid \forall i = 1, \dots, n, \langle \theta, (x_i, 1) \rangle \leq 0 \right\}.
\end{aligned}
\tag{21}
$$

As the $x_i$ are ordered, note that $(C_0, C_1, -C_1, \dots, C_{n-1}, -C_{n-1}, C_n)$ forms a partition of $\mathbb{R}^2$. To prove Lemma 1, it remains to show that any minimiser $(\mu_a, b)$ of Equation (20) has a function $f_{\mu,a,b}$ of the form

$$f_{\mu,a,b}(x) = \tilde{a}x + \tilde{b} + \sum_{i=1}^{n-1} \tilde{a}_i \sigma(\langle \theta_i, (x, 1) \rangle) \quad \text{where } \theta_i \in C_i.$$

Let $f$ be a minimiser of Equation (4). Let $\mu, a, b$ be a minimiser of Equation (20) such that $f_{\mu,a,b} = f$. Define $\tilde{\mu}, \tilde{a}, \tilde{b}$ as

$$
\mathrm{d}\tilde{\mu}(\theta) = \begin{cases} \mathrm{d}\mu(\theta) + \mathrm{d}\mu(-\theta) & \text{for } \theta \in C_i \text{ for any } i = 1, \dots, n-1 \\ 0 & \text{for } \theta \in -C_i \text{ for any } i = 1, \dots, n-1 \\ \mathrm{d}\mu(\theta) & \text{otherwise,} \end{cases}
$$

$$\tilde{a} = a - \sum_{j=1}^{n-1} \int_{-C_j} \theta_1 \mathrm{d}\mu(\theta),$$

$$\tilde{b} = b - \sum_{j=1}^{n-1} \int_{-C_j} \theta_2 \mathrm{d}\mu(\theta).$$

Thanks to the identity $\sigma(u) - u = \sigma(-u)$, $f_{\mu,a,b} = f_{\tilde{\mu},\tilde{a},\tilde{b}}$. Moreover, $\|\tilde{\mu}\|_{\mathrm{TV}} \leq \|\mu\|_{\mathrm{TV}}$, so we can assume w.l.o.g. that the support of $\mu$ is included[10] in $\bigcup_{i=0}^{n} C_i$. In that case, for any $i =, 1 \dots, n$

$$
\begin{aligned}
f(x_i) &= ax_i + b + \sum_{j=0}^{i-1} \int_{C_j} \langle \theta, (x_i, 1) \rangle \mathrm{d}\mu(\theta) \\
&= ax_i + b + \sum_{j=0}^{i-1} \langle \int_{C_j} \theta \mathrm{d}\mu(\theta), (x_i, 1) \rangle.
\end{aligned}
\tag{22}
$$

---

[9]To see that, we can first consider the difference $f_{\mu_j,a_j,b_j}(x_1) - f_{\mu_j,a_j,b_j}(x_2)$ to show that $(a_j)_j$ is bounded. This then leads to the boundedness of $(b_j)_j$ when considering $f_{\mu_j,a_j,b_j}(x_1)$.

[10]We here transform the triple $(\mu, a, b)$, but the corresponding function $f$ remains unchanged.

First, the reduction

$$\tilde{a} = a + \int_{C_0} \theta_1 \mathrm{d}\mu(\theta)$$

$$\tilde{b} = b + \int_{C_0} \theta_2 \mathrm{d}\mu(\theta)$$

$$\tilde{\mu} = \mu_{|\bigcup_{i=1}^{n-1} C_i},$$

does not increase the total variation of $\mu$ and still interpolates the data. As a consequence, the support of $\mu$ is included in $\bigcup_{i=1}^{n-1} C_i$. Now let $\mu = \mu_+ - \mu_-$ be the Jordan decomposition of $\mu$ and define for any $i \in [n-1]$

$$\alpha_i = \int_{C_i} \theta \mathrm{d}\mu_+(\theta) \quad \text{and} \quad \beta_i = \int_{C_i} \theta \mathrm{d}\mu_-(\theta).$$

Note that $\alpha_i$ and $\beta_i$ are both in the positive convex cone $C_i$. For $\theta_i \coloneqq \alpha_i - \beta_i$, Equation (22) rewrites

$$f(x_i) = ax_i + b + \sum_{j=1}^{i-1} \langle \theta_i, (x_i, 1) \rangle.$$

If $\theta_i \in \overline{C}_i \cup \overline{-C}_i$, we can then define

$$\tilde{\mu} = \mu - \mu_{|C_i} + \|\theta_i\| \delta_{\frac{\theta_i}{\|\theta_i\|}}.$$

Thanks to Equation (22), the function $f_{\tilde{\mu},a,b}$ still interpolates the data and

$$\|\tilde{\mu}\|_{\mathrm{TV}} \le \|\mu\|_{\mathrm{TV}} - \|\mu_{|C_i}\|_{\mathrm{TV}} + \|\theta_i\|.$$

By minimisation of $\|\mu\|_{\mathrm{TV}}$, this is an equality. Moreover as $\mu$ is a measure on the sphere,

$$\|\mu_{|C_i}\|_{\mathrm{TV}} = \int_{C_i} \|\theta\| \mathrm{d}\mu_+(\theta) + \int_{C_i} \|\theta\| \mathrm{d}\mu_-(\theta)$$

$$\ge \left\| \int_{C_i} \theta \mathrm{d}\mu_+(\theta) \right\| + \left\| \int_{C_i} \theta \mathrm{d}\mu_-(\theta) \right\|$$

$$= \|\alpha_i\| + \|\beta_i\| \ge \|\theta_i\|.$$

By minimisation, all inequalities are equalities. Jensen's case of equality implies for the first inequality that both $\mu_{+|C_i}$ and $\mu_{-|C_i}$ are Diracs, while the second inequality implies that either $\alpha_i = 0$ or $\beta_i = 0$. Overall, $\mu_{|C_i}$ is at most a single Dirac.

Now if $\theta_i \notin \overline{C}_i \cup \overline{-C}_i$, assume first that $\langle \theta_i, (x_{i+1}, 1) \rangle > 0$. This implies $\langle \theta_i, (x_i, 1) \rangle > 0$ since $\theta_i \notin \overline{C}_i \cup \overline{-C}_i$. This then implies that either $\alpha_i \in \mathring{C}_i$ or $\beta_i \in \mathring{C}_i$, depending on whether $i \ge i_0$ or $i < i_0$. Assume first that $\beta_i \in \mathring{C}_i$ ($i \ge i_0$) and define

$$t = \sup \left\{ t' \in [0,1] \mid t'\alpha_i - \beta_i \in \overline{-C}_i \right\}.$$

By continuity, $t\alpha_i - \beta_i \in \overline{-C}_i$. Moreover $0 < t < 1$, since $\beta_i \in \mathring{C}_i$ and $\theta_i \notin \overline{-C}_i$. We now define

$$\tilde{\mu} = \mu - \mu_{|C_i} + (1-t)\|\alpha_i\| \delta_{\frac{\alpha_i}{\|\alpha_i\|}} + \|t\alpha_i - \beta_i\| \delta_{\frac{t\alpha_i - \beta_i}{\|t\alpha_i - \beta_i\|}}.$$

The function $f_{\tilde{\mu},a,b}$ still interpolates the data. Similarly to the case $\theta_i \in \overline{C}_i \cup \overline{-C}_i$, the minimisation of $\|\mu\|_{\mathrm{TV}}$ implies that $\mu_{|C_i}$ is at most a single Dirac.

If $\alpha_i \in \mathring{C}_i$ instead, similar arguments follow defining

$$t = \sup \left\{ t' \in [0,1] \mid \alpha_i - t'\beta_i \in \overline{C}_i \right\}.$$

Symmetric arguments also hold if $\langle \theta_i, (x_{i+1}, 1) \rangle < 0$. In any case, $\mu_{|C_i}$ is at most a single Dirac. This holds for any $i = 1, \ldots, n-1$, which finally leads to Lemma 1.

### D.2  Proof of Lemma 2

Before proving Lemma 2, let us show a one to one mapping from the parameterisation given by Lemma 1 to a parameterisation given by the sequences of slopes in the points $x_i$. Let us define the

sets

$$\mathcal{S} = \left\{ (s_1, \ldots, s_n) \in \mathbb{R}^n \mid \forall i = 1, \ldots, i_0 - 2, s_i \in S_i(s_{i+1}), \right.$$

$$\left. (s_{i_0-1}, s_{i_0}) \in \Lambda \quad \text{and} \quad \forall i = i_0, \ldots, n-1, s_{i+1} \in S_i(s_i) \right\} \tag{23}$$

and

$$\mathcal{I} = \left\{ (a, b, (a_i, \tau_i)_{i=1,\ldots,n-1}) \mid \forall j = 1, \ldots, n, \ a x_j + b + \sum_{i=1}^{n-1} a_i (x_j - \tau_i)_+ = y_j, \quad \tau_{i_0-1} \in (x_{i_0-1}, x_{i_0}), \right.$$

$$\tau_i = \frac{x_i + x_{i+1}}{2} \text{ if } a_i = 0, \quad \forall i \in \{1, \ldots, i_0 - 2\}, \tau_i \in (x_i, x_{i+1}]$$

$$\left. \text{and} \quad \forall i \in \{i_0, \ldots, n-1\}, \tau_i \in [x_i, x_{i+1}) \right\}.$$

The condition $\tau_i = \frac{x_i + x_{i+1}}{2}$ if $a_i = 0$ in the definition of $\mathcal{I}$ is just to avoid redundancy, as any arbitrary value of $\tau_i$ would yield the same interpolating function. Lemma 6 below gives a one to one mapping between these two sets.

**Lemma 6.** *The function*

$$\psi : \begin{array}{c} \mathcal{I} \to \mathcal{S} \\ (a_0, b_0, (a_i, \tau_i)_{i=1,\ldots,n-1}) \mapsto (\sum_{j=0}^{i-1} a_j)_{i=1,\ldots,n-1} \end{array}$$

*is a one to one mapping. Its inverse is given by*

$$\psi^{-1} : \begin{array}{c} \mathcal{S} \to \mathcal{I} \\ (s_i)_{i \in [n]} \mapsto (a_0, b_0, (a_i, \tau_i)_{i \in [n-1]}) \end{array}$$

*where*

$$a_0 = s_1; \qquad b_0 = y_1 - s_1 x_1; \qquad a_i = s_{i+1} - s_i \quad \text{for any } i \in [n-1];$$

$$\tau_i = \begin{cases} \frac{s_{i+1} - \delta_i}{s_{i+1} - s_i} x_{i+1} + \frac{\delta_i - s_i}{s_{i+1} - s_i} x_i & \text{if } s_{i+1} \neq s_i \\ \frac{x_i + x_{i+1}}{2} & \text{otherwise} \end{cases}.$$

*Proof.* For $(a_0, b_0, (a_i, \tau_i)_{i=1,\ldots,n-1}) \in \mathcal{I}$, let $f$ be the associated interpolator:

$$f(x) = a_0 x + b_0 + \sum_{i=1}^{n-1} a_i (x - \tau_i)_+$$

and let $(s_i)_{i \in [n]} = \psi(a_0, b_0, (a_i, \tau_i)_i)$. Given the definition of $\psi$, it is straightforward to check that $s_i$ corresponds to the left (resp. right) derivative of $f$ at $x_i \geq 0$ (resp. $x_i < 0$). We actually have the two following inequalities linking the parameters $(a_0, b_0, (a_i, \tau_i)_i)$ and $(s_i)_i$ for any $i \in [n-1]$:

$$s_i + a_i = s_{i+1},$$
$$y_i + s_i (x_{i+1} - x_i) + a_i (x_{i+1} - \tau_i) = y_{i+1}.$$

The first equality comes from the (left or right) derivatives of $f$ in $x_i$, while the second equality is due to the interpolation of the data by $f$. These two equalities imply that an interpolator with ReLU parameters in $\mathcal{I}$ (i.e., $f$) can be equivalently described by its (left or right) derivatives in each $x_i$. A straightforward computation then allows to show that $\psi$ and $\psi^{-1}$ are well defined and indeed verify $\psi \circ \psi^{-1} = I_{\mathcal{S}}$ and $\psi^{-1} \circ \psi = I_{\mathcal{I}}$. $\square$

Using this bijection from $\mathcal{I}$ to $\mathcal{S}$, we can now prove Lemma 2. Note for the remaining of the proof $\alpha = \min_{\substack{f \\ \forall i \in [n], f(x_i) = y_i}} \int_{\mathbb{R}} \sqrt{1 + x^2} \mathrm{d}|f''(x)|$. Thanks to Lemma 1, we have the first equivalence:

$$\alpha = \min_{(a_0, b_0, (a_i, \tau_i)_{i=1,\ldots,n-1}) \in \mathcal{I}} \sum_{i=1}^{n-1} |a_i| \sqrt{1 + \tau_i^2}.$$

For any $(a_0, b_0, (a_i, \tau_i)_{i=1,\ldots,n-1}) \in \mathcal{I}$, we can define thanks to Lemma 6 $(s_i)_i = \psi(a_0, b_0, (a_i, \tau_i)_i) \in \mathcal{S}$. We then have $(a_0, b_0, (a_i, \tau_i)_i) = \psi^{-1}((s_i)_i)$. Moreover, by definition

of $\psi^{-1}$, we can easily check that

$$|a_i|\sqrt{1+\tau_i^2} = \sqrt{a_i^2 + (a_i\tau_i)^2}$$
$$= \sqrt{(s_{i+1}-s_i)^2 + ((s_{i+1}-\delta_i)x_{i+1} + (s_i-\delta_i)x_i)^2}$$
$$= g_{i+1}(s_{i+1}, s_i). \tag{24}$$

As $\psi$ is a one to one mapping, we have for any function $h$ the equivalence $\min_{u\in\psi^{-1}(\mathcal{S})} h(u) = \min_{s\in\mathcal{S}} h(\psi^{-1}(s))$. In particular, thanks to Equation (24):

$$\min_{(a_0,b_0,(a_i,\tau_i)_{i=1,\dots,n-1})\in\mathcal{I}} \sum_{i=1}^{n-1} |a_i|\sqrt{1+\tau_i^2} = \min_{(s_i)_i\in\mathcal{S}} \sum_{i=1}^{n-1} g_{i+1}(s_{i+1}, s_i). \tag{25}$$

From there, define for any $i \geq i_0$,

$$d_i(s_i) = \min_{\substack{(\tilde{s})_j\in\mathcal{S} \\ \text{s.t. } \tilde{s}_i=s_i}} \sum_{j=i}^{n-1} g_{j+1}(\tilde{s}_{j+1}, \tilde{s}_j);$$

and for any $i < i_0$

$$d_i(s_i) = \min_{\substack{(\tilde{s})_j\in\mathcal{S} \\ \text{s.t. } \tilde{s}_i=s_i}} \sum_{j=1}^{i-1} g_{j+1}(\tilde{s}_{j+1}, \tilde{s}_j).$$

Obviously, we have from Equation (25) and the definition of $\mathcal{S}$ that

$$\alpha = \min_{(s_{i_0-1},s_{i_0})\in\Lambda} g_{i_0}(s_{i_0}, s_{i_0-1}) + d_{i_0-1}(s_{i_0-1}) + d_{i_0}(s_{i_0}). \tag{26}$$

It now remains to show by induction that for any $i$ that $c_i = d_i$. This is obviously the case for $i = n$. Let us now consider $i \in \{i_0, \dots, n-1\}$. The definition of $d_i$ leads to

$$d_i(s_i) = \min_{\substack{(\tilde{s})_j\in\mathcal{S} \\ \text{s.t. } \tilde{s}_i=s_i}} \sum_{j=i}^{n-1} g_{j+1}(\tilde{s}_{j+1}, \tilde{s}_j)$$

$$= \min_{s_{i+1}\in S_i(s_i)} \min_{\substack{(\tilde{s})_j\in\mathcal{S} \\ \text{s.t. } \tilde{s}_i=s_i \\ \tilde{s}_{i+1}=s_{i+1}}} g_{i+1}(s_{i+1}, s_i) + \sum_{j=i+1}^{n-1} g_{j+1}(\tilde{s}_{j+1}, \tilde{s}_j)$$

$$= \min_{s_{i+1}\in S_i(s_i)} g_{i+1}(s_{i+1}, s_i) + \min_{\substack{(\tilde{s})_j\in\mathcal{S} \\ \text{s.t. } \tilde{s}_i=s_i \\ \tilde{s}_{i+1}=s_{i+1}}} \sum_{j=i+1}^{n-1} g_{j+1}(\tilde{s}_{j+1}, \tilde{s}_j). \tag{27}$$

Now note that for any $s_{i+1} \in S_i(s_i)$, we have the equality of the sets

$$\left\{ (\tilde{s}_j)_{j\geq i+1} \mid (\tilde{s})_{j\in[n-1]} \in \mathcal{S} \text{ s.t. } \tilde{s}_i = s_i \text{ and } \tilde{s}_{i+1} = s_{i+1} \right\} = \left\{ (\tilde{s}_j)_{j\geq i+1} \mid (\tilde{s})_{j\in[n-1]} \in \mathcal{S} \text{ s.t. } \tilde{s}_{i+1} = s_{i+1} \right\}$$

Since the last term in Equation (27) only depends on $(\tilde{s}_j)_{j\geq i+1}$, this implies that

$$d_i(s_i) = \min_{s_{i+1}\in S_i(s_i)} g_{i+1}(s_{i+1}, s_i) + \min_{\substack{(\tilde{s})_j\in\mathcal{S} \\ \text{s.t. } \tilde{s}_{i+1}=s_{i+1}}} \sum_{j=i+1}^{n-1} g_{j+1}(\tilde{s}_{j+1}, \tilde{s}_j)$$

$$= \min_{s_{i+1}\in S_i(s_i)} g_{i+1}(s_{i+1}, s_i) + d_{i+1}(s_{i+1}).$$

By induction, it naturally comes from the definition of $c_i$ that $c_i = d_i$ for any $i \geq i_0$. Symmetric arguments hold for any $i < i_0$, which finally gives $c_i = d_i$ for any $i \in [n]$. Equation (26) then yields Lemma 2.

# E    Proof of Section 5

The proofs of this section are shown in the case where $x_1 < 0$ and $x_n \geq 0$. When all the $x$ are positive, i.e., $x_1 \geq 0$, the adapted version of Lemma 2 would yield for $i_0 = 1$ the equivalence[11]

$$\min_{\substack{f \\ \forall i \in [n], f(x_i) = y_i}} \int_{\mathbb{R}} \sqrt{1 + x^2} \mathrm{d}|f''(x)| = \min_{s_{i_0} \in \mathbb{R}} c_{i_0}(s_{i_0}).$$

The proofs of Appendix E can then be easily adapted to this case (and similarly if $x_n < 0$). Appendix E.5 at the end of the section more precisely states how to adapt them to this case.

## E.1    Proof of Theorem 2

Before proving Theorem 2, Lemma 7 below provides important properties verified by the functions $c_i$ defined in Equation (8).

**Lemma 7.** *For each $i \in \{i_0, \dots, n-1\}$, the function $c_i$ is convex, $\sqrt{1 + x_i^2}$-Lipschitz on $\mathbb{R}$ and minimised for $s_i = \delta_i$.*
*Moreover, on both intervals $(-\infty, \delta_i]$ and $[\delta_i, +\infty)$:*

1. *either $c_i(s_i) = \sqrt{1 + x_i^2}|s_i - \delta_i| + c_{i+1}(\delta_i)$ for all $s_i$ in the considered interval, or $c_i$ is strictly convex on the considered interval;*

2. *$|c_i(s_i) - c_i(s_i')| \geq \frac{1 + x_i x_{i+1}}{\sqrt{1 + x_{i+1}^2}}|s_i - s_i'|$ for all $s_i, s_i'$ in the considered interval.*

*Similarly, for each $i \in \{1, \dots, i_0 - 2\}$, the function $c_{i+1}$ is convex, $\sqrt{1 + x_{i+1}^2}$-Lipschitz on $\mathbb{R}$ and and minimised for $s_{i+1} = \delta_i$.*
*Moreover, on both intervals $(-\infty, \delta_i]$ and $[\delta_i, +\infty)$:*

1. *either $c_{i+1}(s_{i+1}) = \sqrt{1 + x_{i+1}^2}|s_{i+1} - \delta_i| + c_i(\delta_i)$ for all $s_{i+1}$ in the considered interval, or $c_{i+1}$ is strictly convex on the considered interval;*

2. *$|c_{i+1}(s_{i+1}) - c_{i+1}(s_{i+1}')| \geq \frac{1 + x_i x_{i+1}}{\sqrt{1 + x_i^2}}|s_{i+1} - s_{i+1}'|$ for all $s_{i+1}, s_{i+1}'$ in the considered interval.*

*Proof.* For any $i \in \{1, \dots, i_0 - 2\}$, we prove the result by (backward) induction. Since $c_n = 0$, a straightforward calculation gives[12]

$$c_{n-1}(s_{n-1}) = \sqrt{1 + x_{n-1}^2}|s_{n-1} - \delta_{n-1}|,$$

which gives the wanted properties for $i = n - 1$.

Now consider $i \in \{i_0, \dots, n-2\}$ such that $c_{i+1}$ verifies all the properties in the first part of Lemma 7. We first show the Lipschitz property of $c_i$. Let $s_i, s_i' < \delta_i$ first. By inductive assumption, the function $s_{i+1} \mapsto g_{i+1}(s_{i+1}, s_i) + c_{i+1}(s_{i+1})$ reaches a minimum on $[\delta_i, +\infty)$. Consider $s_{i+1} \geq \delta_i$ such that

$$c_i(s_i) = g_{i+1}(s_{i+1}, s_i) + c_{i+1}(s_{i+1}).$$

Also by minimisation, $c_i(s_i') \leq g_{i+1}(s_{i+1}, s_i') + c_{i+1}(s_{i+1})$. For the vectors $u = (x_{i+1}, 1)$ and $v = (x_i, 1)$, it then holds:

$$c_i(s_i') - c_i(s_i) \leq g_{i+1}(s_{i+1}, s_i') - g_{i+1}(s_{i+1}, s_i)$$
$$= \|(s_{i+1} - \delta_i)u - (s_i' - \delta_i)v\| - \|(s_{i+1} - \delta_i)u - (s_i - \delta_i)v\|$$
$$\leq \|(s_i - s_i')v\| = \sqrt{1 + x_i^2}|s_i - s_i'|.$$

The first equality comes from the definition of $g_{i+1}$ as a norm and the second inequality comes from the triangle inequality. By symmetry, we showed $|c_i(s_i') - c_i(s_i)| \leq \sqrt{1 + x_i^2}|s_i - s_i'|$ for $s_i, s_i' < \delta_i$.

---

[11]Note that in that case $c_1 \not\equiv 0$. Instead, $c_1$ is defined through the recursion given in Equation (8).

[12]This calculation uses the fact that both $x_{n-1}$ and $x_n$ are positive, which implies that the minimal $s_n$ in the definition of $c_{n-1}$ is $\delta_{n-1}$

Note that if $s_i = \delta_i$, then $s_{i+1} = \delta_i$ and we show similarly that $c_i(s_i') - c_i(\delta_i) \leq \sqrt{1 + x_i^2}|\delta_i - s_i'|$. Moreover,

$$c_i(s_i') - c_i(\delta_i) = \min_{s_{i+1}' \geq \delta_i} \|(s_{i+1}' - \delta_i)u - (s_i' - \delta_i)v\| + c_{i+1}(s_{i+1}') - c_{i+1}(\delta_i)$$

$$\geq \min_{s_{i+1}' \geq \delta_i} \|(s_{i+1}' - \delta_i)u - (s_i' - \delta_i)v\| - \|(s_{i+1}' - \delta_i)u\|$$

$$\geq 0.$$

The first inequality comes from the Lipschitz property of $c_{i+1}$. The second from the fact that $(s_{i+1}' - \delta_i)u$ and $(s_i' - \delta_i)v$ are negatively correlated, since $x_i$ and $x_{i+1}$ are both positive. As a consequence, $c_i$ is $\sqrt{1 + x_i^2}$-Lipschitz on $(-\infty, \delta_i]$. Symmetrically, it is also $\sqrt{1 + x_i^2}$-Lipschitz on $[\delta_i, +\infty)$, which finally implies it is $\sqrt{1 + x_i^2}$-Lipschitz on $\mathbb{R}$. Moreover, the last calculation also shows that $c_i$ is minimised for $s_i = \delta_i$.

Let us now show that $c_i$ verifies the first point on $(-\infty, \delta_i]$. By continuity, we only have to show it on $(-\infty, \delta_i)$. Let $s_i \in (-\infty, \delta_i)$, we then have by definition

$$c_i(s_i) = \min_{s_{i+1} \geq \delta_i} g_{i+1}(s_{i+1}, s_i) + c_{i+1}(s_{i+1}).$$

If $\delta_{i+1} \leq \delta_i$, note that both functions $g_{i+1}(\cdot, s_i)$ and $c_{i+1}$ are increasing on $[\delta_i, +\infty)$[13]. The minimum is thus reached for $s_{i+1} = \delta_i$ and

$$c_i(s_i) = \sqrt{1 + x_i^2}|s_i - \delta_i| + c_{i+1}(\delta_i).$$

If $\delta_{i+1} > \delta_i$, both functions $g_{i+1}(\cdot, s_i)$ and $c_{i+1}$ are increasing on $[\delta_{i+1}, +\infty)$. As a consequence, we can then rewrite

$$c_i(s_i) = \min_{s_{i+1} \in [\delta_i, \delta_{i+1}]} g_{i+1}(s_{i+1}, s_i) + c_{i+1}(s_{i+1}). \tag{28}$$

Assume first that $c_{i+1}(s_{i+1}) = \sqrt{1 + x_{i+1}^2}|s_{i+1} - \delta_{i+1}| + c_{i+2}(\delta_{i+1})$ on $[\delta_i, \delta_{i+1}]$. By triangle inequality, we actually have

$$g_{i+1}(s_{i+1}, s_i) \geq g_{i+1}(\delta_{i+1}, s_i) - \sqrt{1 + x_{i+1}^2}|\delta_{i+1} - s_{i+1}|.$$

This leads for $s_{i+1} \in [\delta_i, \delta_{i+1}]$ to

$$g_{i+1}(s_{i+1}, s_i) + c_{i+1}(s_{i+1}) \geq g_{i+1}(\delta_{i+1}, s_i) + c_{i+2}(\delta_{i+1}).$$

The minimum in Equation (28) is thus reached for $s_{i+1} = \delta_{i+1}$, which finally gives for any $s_i \leq \delta_i$

$$c_i(s_i) = g_{i+1}(\delta_{i+1}, s_i) + c_{i+2}(\delta_{i+1}).$$

Since $\delta_{i+1} > \delta_i$, it is easy to check that $g_{i+1}(\delta_{i+1}, \cdot)$ is strictly convex on $(-\infty, \delta_i)$ and so is $c_i$.

Let us now assume the last case, where $c_{i+1}$ is strictly convex on $[\delta_i, \delta_{i+1}]$. By contradiction, assume that the first point on $(-\infty, \delta_i]$ does not hold. Note in the following $h(s_{i+1}, s_i) = g_{i+1}(s_{i+1}, s_i) + c_{i+1}(s_{i+1})$. For $s_i, s_i' < \delta_i$, by continuity of $h$, let $s_{i+1}, s_{i+1}' \in [\delta_i, \delta_{i+1}]$ be such that

$$c_i(s_i) = h(s_{i+1}, s_i) \quad \text{and} \quad c_i(s_i') = h(s_{i+1}', s_i').$$

For any $t \in (0, 1)$, by convexity of $h$:

$$c_i(ts_i + (1 - t)s_i') \leq h(t(s_{i+1}, s_i) + (1 - t)(s_{i+1}, s_i))$$

$$\leq th(s_{i+1}, s_i) + (1 - t)h(s_{i+1}', s_i')$$

$$= tc_i(s_i) + (1 - t)c_i(s_i').$$

$c_i$ is thus convex on $(-\infty, \delta_i]$. Moreover, the case of equality corresponds to the case of equality for both $g_{i+1}$ and $c_{i+1}$:

$$g_{i+1}(t(s_{i+1}, s_i) + (1 - t)(s_{i+1}, s_i)) = tg_{i+1}(s_{i+1}, s_i) + (1 - t)g_{i+1}(s_{i+1}', s_i')$$

$$c_{i+1}(ts_{i+1} + (1 - t)s_{i+1}') = tc_{i+1}(s_{i+1}) + (1 - t)c_{i+1}(s_{i+1}').$$

The former leads to the colinearity of the vectors $(s_{i+1} - \delta_i, s_i - \delta_i)$ and $(s_{i+1}' - \delta_i, s_i' - \delta_i)$; the latter gives $s_{i+1} = s_{i+1}'$ by strict convexity of $c_{i+1}$. Two cases are then possible

$$\begin{cases} \text{either } s_{i+1} = \delta_i = s_{i+1}' \\ \text{or } s_i = s_i'. \end{cases}$$

---

[13]Here again, we use the fact that $x_i$ and $x_{i+1}$ are positive.

The former case then implies that $c_i(s_i) = \sqrt{1 + x_i^2}|s_i - \delta_i| + c_{i+1}(\delta_i)$. Since $c_i(\delta_i) = c_{i+1}(\delta_i)$, $c_i$ is $\sqrt{1 + x_i^2}$-Lipschitz and convex on $(-\infty, \delta_i]$, this leads to $c_i(s) = \sqrt{1 + x_i^2}|s - \delta_i| + c_{i+1}(\delta_i)$ for any $s \in (-\infty, \delta_i]$. This contradicts the assumption that the first point does not hold on $(-\infty, \delta_i]$. Necessarily, we have $s_i = s_i'$. So $c_i$ is strictly convex on $(-\infty, \delta_i]$, which leads to another contradiction: the first point does hold on $(-\infty, \delta_i]$.

Finally, we just showed that in any case, $c_i$ is either strictly convex or equal to $s_i \mapsto \sqrt{1 + x_i^2}|s_i - \delta_i|$ on $(-\infty, \delta_i]$. Symmetric arguments yield the same on $[\delta_i, +\infty)$. $c_i$ is thus minimised in $\delta_i$, $\sqrt{1 + x_i^2}$-Lipschitz and verifies the first point on both intervals $(-\infty, \delta_i]$ and $[\delta_i, +\infty)$. This directly implies that $c_i$ is convex on $\mathbb{R}$.

It now remains to show the second point on the two intervals. Let us show it on $(-\infty, \delta_i]$: on $(-\infty, \delta_i)$ is actually sufficient by continuity. Consider $s_i < s_i' < \delta_i$ and $s_{i+1} \in [\delta_i, +\infty)$ such that

$$c_i(s_i) = g_{i+1}(s_{i+1}, s_i) + c_{i+1}(s_{i+1}).$$

By definition of $c_i$,

$$c_i(s_i) - c_i(s_i') \geq g_{i+1}(s_{i+1}, s_i) - g_{i+1}(s_{i+1}, s_i').$$

Straightforward computations yield that the function

$$h_2 : \begin{array}{c} (-\infty, \delta_i] \to \mathbb{R}_+ \\ s \mapsto g_{i+1}(s_{i+1}, s) \end{array}$$

is convex and $h_2'(\delta_i) = -\frac{1 + x_i x_{i+1}}{\sqrt{1 + x_{i+1}^2}}$. Thus, $h_2' \leq -\frac{1 + x_i x_{i+1}}{\sqrt{1 + x_{i+1}^2}}$, which finally implies

$$c_i(s_i) - c_i(s_i') \geq h(s_i) - h(s_i')$$

$$\geq \frac{1 + x_i x_{i+1}}{\sqrt{1 + x_{i+1}^2}}(s_i' - s_i).$$

The second point is thus verified on $(-\infty, \delta_i)$ and on $(-\infty, \delta_i]$ by continuity. Symmetric arguments lead to the same property on $[\delta_i, +\infty)$.

By induction, this implies the first part of Lemma 7. Symmetric arguments lead to the second part of Lemma 7 for $i \leq i_0 - 2$. $\qquad\square$

We can now prove Theorem 2. Following the proof of Lemma 2, there is a unique minimiser of Equation (4) if and only if the following problem admits a unique minimiser:

$$\min_{\mathbf{s} \in \mathcal{S}} \sum_{i=1}^{n-1} g_{i+1}(s_{i+1}, s_i). \tag{29}$$

We already know that the minimum is attained thanks to Lemma 1. By construction of the functions $c_i$, any minimum $\tilde{\mathbf{s}}$ of Equation (29) verifies

$$\tilde{s}_i \in \underset{s_i \in S_i(\tilde{s}_{i+1})}{\operatorname{argmin}} \; g_{i+1}(\tilde{s}_{i+1}, s_i) + c_i(s_i) \quad \text{for any } i \in [i_0 - 2] \tag{30}$$

$$(\tilde{s}_{i_0-1}, \tilde{s}_{i_0}) \in \underset{(s_{i_0-1}, s_{i_0}) \in \Lambda}{\operatorname{argmin}} \; g_{i_0}(s_{i_0}, s_{i_0-1}) + c_{i_0-1}(s_{i_0-1}) + c_{i_0}(s_{i_0}) \tag{31}$$

$$\tilde{s}_{i+1} \in \underset{s_{i+1} \in S_i(\tilde{s}_i)}{\operatorname{argmin}} \; g_{i+1}(s_{i+1}, \tilde{s}_i) + c_{i+1}(s_{i+1}) \quad \text{for any } i \in \{i_0, \ldots, n-1\}$$

It now remains to show that all these problems admit unique minimisers. First assume Equation (31) admits different minimisers $(s_{i_0-1}, s_{i_0})$ and $(s_{i_0-1}', s_{i_0}')$. Note in the following $h_{i_0-1} : (s, s') \mapsto g_{i_0}(s, s') + c_{i_0-1}(s') + c_{i_0}(s)$. By minimisation and convexity of the three functions $g_{i_0}, c_{i_0-1}, c_{i_0}$, for any $t \in (0, 1)$:

$$h_{i_0-1}(t(s_{i_0-1}, s_{i_0}) + (1-t)(s_{i_0-1}', s_{i_0}')) \leq t h_{i_0-1}(s_{i_0-1}, s_{i_0}) + (1-t) h_{i_0-1}(s_{i_0-1}', s_{i_0}') \tag{32}$$

$$= h_{i_0-1}(s_{i_0-1}, s_{i_0}). \tag{33}$$

The whole segment joining $(s_{i_0-1}, s_{i_0})$ and $(s_{i_0-1}', s_{i_0}')$ is then a minimiser. Without loss of generality, we can thus assume that both $s_{i_0}$ and $s_{i_0-1}'$ are on the same side of $\delta_{i_0-1}$ (e.g. smaller than $\delta_{i_0-1}$) and both $s_{i_0}$ and $s_{i_0}'$ are on the same side of $\delta_{i_0}$.

Moreover, Equation (33) implies an equality on $g_{i_0}$ that leads to the colinearity of the vectors $(s_{i_0-1} - \delta_{i_0-1}, s_{i_0} - \delta_{i_0-1})$ and $(s_{i_0-1}' - \delta_{i_0-1}, s_{i_0}' - \delta_{i_0-1})$. In particular, both $s_{i_0} \neq s_{i_0}'$ and

$s_{i_0-1} \neq s'_{i_0-1}$. Moreover, we have equality cases on both $c_{i_0-1}$ and $c_{i_0}$ implying, thanks to the first point of Lemma 7

$$|c_{i_0-1}(s_{i_0-1}) - c_{i_0-1}(s'_{i_0-1})| = \sqrt{1+x_{i_0-1}^2}|s_{i_0-1} - s_{i_0-1}|$$
$$|c_{i_0}(s_{i_0}) - c_{i_0}(s'_{i_0})| = \sqrt{1+x_{i_0}^2}|s_{i_0} - s_{i_0}|. \tag{34}$$

For $u = (x_{i_0-1}, 1)$ and $v = (x_{i_0}, 1)$, we have by (positive) colinearity of $(s_{i_0-1} - \delta_{i_0-1}, s_{i_0} - \delta_{i_0-1})$ and $(s'_{i_0-1} - \delta_{i_0-1}, s'_{i_0} - \delta_{i_0-1})$:

$$|g_{i_0}(s_{i_0}, s_{i_0-1}) - g_{i_0}(s'_{i_0}, s'_{i_0-1})| = \left| \|(s_{i_0} - \delta_{i_0-1})v - (s_{i_0-1} - \delta_{i_0-1})u\| - \|(s'_{i_0} - \delta_{i_0-1})v - (s'_{i_0-1} - \delta_{i_0-1})u\| \right|$$
$$= \|(s_{i_0} - s'_{i_0})v - (s_{i_0-1} - s'_{i_0-1})u\|.$$

Since $s_{i_0} \neq s'_{i_0}$ and $s_{i_0-1} \neq s'_{i_0-1}$, the triangle inequality gives both strict inequalities

$$\left| \sqrt{1+x_{i_0}^2}|s_{i_0} - s'_{i_0}| - \sqrt{1+x_{i_0-1}^2}|s_{i_0-1} - s'_{i_0-1}| \right| < |g_{i_0}(s_{i_0}, s_{i_0-1}) - g_{i_0}(s'_{i_0}, s'_{i_0-1})|,$$

$$\sqrt{1+x_{i_0}^2}|s_{i_0} - s'_{i_0}| + \sqrt{1+x_{i_0-1}^2}|s_{i_0-1} - s'_{i_0-1}| > |g_{i_0}(s_{i_0}, s_{i_0-1}) - g_{i_0}(s'_{i_0}, s'_{i_0-1})|.$$

Using this with Equation (34), this yields

$$g_{i_0}(s_{i_0}, s_{i_0-1}) - g_{i_0}(s'_{i_0}, s'_{i_0-1}) \neq c_{i_0}(s_{i_0}) - c_{i_0}(s'_{i_0}) + c_{i_0-1}(s_{i_0-1}) - c_{i_0-1}(s'_{i_0-1}).$$

This contradicts the fact that $(s_{i_0-1}, s_{i_0})$ and $(s'_{i_0-1}, s'_{i_0})$ both minimise Equation (31). Hence, Equation (31) admits a unique minimiser.

Also the minimisation problem

$$\min_{s_{i+1} \in S_i(\tilde{s}_i)} g_{i+1}(s_{i+1}, \tilde{s}_i) + c_{i+1}(s_{i+1})$$

admits a unique minimiser for any $i \in \{i_0, \dots, n-1\}$. Indeed, either $\tilde{s}_i = \delta_i$ in which case the constraint set is a singleton, or the function $s_{i+1} \mapsto g_{i+1}(s_{i+1}, \tilde{s}_i)$ is strictly convex for $\tilde{s}_i \neq \delta_i$. A symmetric argument exists for the minimisation problem of Equation (28). It thus concludes the proof of Theorem 2.

### E.2 Proof of Lemma 3

Let $i \in \{i_0, \dots, n-1\}$. Recall that

$$s_{i+1}^* = \underset{s_{i+1} \in S_i(s_i^*)}{\operatorname{argmin}} g_{i+1}(s_{i+1}, s_i^*) + c_{i+1}(s_{i+1}).$$

If $i = n-1$, the objective is obviously minimised for $s_n^* = \delta_{n-1}$ as both $x_{n-1}$ and $x_n$ are positive. Otherwise, assume for example that $s_i^* > \delta_i$. Thanks to Lemma 7, the objective is decreasing on $(-\infty, \min(\delta_i, \delta_{i+1})]$ and $S_i(s_i^*) = [\delta_i, +\infty)$ which yields that $s_{i+1}^* \in [\min(\delta_i, \delta_{i+1}), \delta_i] \subset [\min(\delta_i, \delta_{i+1}), \max(\delta_i, \delta_{i+1})]$. The case $s_i^* = \delta_i$ is trivial and similar arguments hold for $s_i^* < \delta_i$.

Now consider $i = i_0 - 1$. Assume first that $s_{i_0-1}^* > \delta_{i_0-1}$, then

$$s_{i_0}^* = \underset{s_{i_0} < \delta_{i_0-1}}{\operatorname{argmin}} g_{i_0}(s_{i_0}, s_{i_0-1}^*) + c_{i_0}(s_{i_0}).$$

Thanks to the last point of Lemma 7

$$c_{i_0}(s_{i_0}) - c_{i_0}(s'_{i_0}) \geq \frac{1 + x_{i_0}x_{i_0+1}}{\sqrt{1+x_{i_0+1}^2}}(s_{i_0} - s'_{i_0}) \qquad \text{for any } s_{i_0} < s'_{i_0} \leq \delta_{i_0}. \tag{35}$$

Note that the function

$$h: \begin{array}{l} (-\infty, \delta_{i_0-1}] \to \mathbb{R}_+ \\ s \mapsto g_{i_0}(s, s_{i_0-1}^*) \end{array}$$

is convex and verifies $h'(\delta_{i_0-1}) = -\frac{1+x_{i_0-1}x_{i_0}}{\sqrt{1+x_{i_0-1}^2}}$. Since $x_{i_0-1} < 0 \leq x_{i_0+1}$, it comes

$$-\frac{1+x_{i_0-1}x_{i_0}}{\sqrt{1+x_{i_0-1}^2}} \leq x_{i_0} \leq \frac{1+x_{i_0}x_{i_0+1}}{\sqrt{1+x_{i_0+1}^2}}.$$

Thanks to Equation (35), the function $s_{i_0} \mapsto g_{i_0}(s_{i_0}, s_{i_0-1}^*) + c_{i_0}(s_{i_0})$ is thus decreasing on $(-\infty, \min(\delta_{i_0-1}, \delta_{i_0})]$. As above, this implies that $s_{i_0}^* \in [\min(\delta_{i_0-1}, \delta_{i_0}), \max(\delta_{i_0-1}, \delta_{i_0})]$. The case $s_{i_0-1}^* = \delta_{i_0-1}$ is trivial and similar arguments hold if $s_{i_0-1}^* < \delta_{i_0-1}$.

We showed $s_{i+1}^* \in [\min(\delta_i, \delta_{i+1}), \max(\delta_i, \delta_{i+1})]$ for any $i \in \{i_0, \ldots, n\}$. Symmetric arguments hold for $i \in [i_0 - 1]$. This concludes the proof of Lemma 3.

### E.3    Proof of Lemma 4

Let us first prove that any sparsest interpolator $f$ has at least a number of kinks given by the right sum. For that, we actually show that for $k \geq 1$, on any interval $(x_{n_k-1}, x_{n_{k+1}})$ with $\delta_{n_k-1} \neq \delta_{n_k}$, $f$ has at least $\left\lceil \frac{n_{k+1}-n_k}{2} \right\rceil$ kinks, whose signs are given by $\text{sign}(\delta_{n_k} - \delta_{n_k-1})$. Consider any $k \geq 1$ such that $\delta_{n_k-1} \neq \delta_{n_k}$. Assume w.l.o.g. that $\delta_{n_k-1} < \delta_{n_k}$. By the definition of Equation (10):
$$\delta_j > \delta_{j-1} \quad \text{for any } j \in \{n_k, \ldots, n_{k+1} - 1\}.$$

Obviously, $f$ must count at least one positive kink on each interval of the form[14] $(x_{j-1}, x_{j+1})$ for any $n_k \leq j \leq n_{k+1} - 1$. Note that we can build $\left\lceil \frac{n_{k+1}-n_k}{2} \right\rceil$ disjoint such intervals. Thus, $f$ has at least $\left\lceil \frac{n_{k+1}-n_k}{2} \right\rceil$ positive kinks on $(x_{n_k-1}, x_{n_{k+1}})$.

The intervals of the form $(x_{n_k-1}, x_{n_{k+1}})$ with $\delta_{n_k-1} < \delta_{n_k}$ are disjoint by definition. As a consequence, $f$ has a total number of positive kinks at least
$$\sum_{k \geq 1} \left\lceil \frac{n_{k+1} - n_k}{2} \right\rceil \mathbb{1}_{\delta_{n_k-1} < \delta_{n_k}}.$$

Similarly, $f$ has a total number of negative kinks at least
$$\sum_{k \geq 1} \left\lceil \frac{n_{k+1} - n_k}{2} \right\rceil \mathbb{1}_{\delta_{n_k-1} > \delta_{n_k}},$$

which leads to the first part of Lemma 4
$$\min_{\substack{f \\ \forall i, f(x_i) = y_i}} \|f''\|_0 \geq \sum_{k \geq 1} \left\lceil \frac{n_{k+1} - n_k}{2} \right\rceil \mathbb{1}_{\delta_{n_k-1} \neq \delta_{n_k}}.$$

We now construct an interpolating function that has exactly the desired number of kinks. Note that the problem considered in Lemma 4 is shift invariant (which is not the case of Equation (4)). As a consequence, we can assume without loss of generality that $x_1 \geq 0$. This simplifies the definition of the following sequence of slopes $\mathbf{s} \in S$:
$$s_1 = \delta_1$$
and for any $i \in \{2, \ldots, n\}$,    $s_i = \begin{cases} \delta_{i-1} \text{ if } (s_{i-1} = \delta_{i-1} \text{ or } i = n_k \text{ for some } k \geq 1) \\ s_i = \delta_i \text{ otherwise.} \end{cases}$

It is easy to check that $\mathbf{s} \in S$. We now consider the function $f$ associated to the sequence of slopes by the mapping of Lemma 6 and an interval $[x_{n_k-1}, x_{n_{k+1}})$ with $\delta_{n_k-1} \neq \delta_{n_k}$. By definition, $s_{n_k+1+2p} = \delta_{n_k+1+2p}$ for any $p$ such that $n_k + 1 \leq n_k + 1 + 2p < n_{k+1}$. This implies that $f$ has no kink in the interval $[x_{n_k+1+2p}, x_{n_k+2+2p})$. From there, a simple calculation shows that $f$ has at most $\left\lceil \frac{n_{k+1}-n_k}{2} \right\rceil$ kinks on $[x_{n_k}, x_{n_{k+1}})$. Moreover, as $s_i = \delta_{i-1}$ if $i = n_k$, $f$ has no kink on intervals $[x_{n_k}, x_{n_{k+1}})$ when $\delta_{n_k-1} = \delta_{n_k}$. $f$ is thus an interpolating function with at most
$$\sum_{k \geq 1} \left\lceil \frac{n_{k+1} - n_k}{2} \right\rceil \mathbb{1}_{\delta_{n_k-1} \neq \delta_{n_k}},$$

kinks, which concludes the proof of Lemma 4.

### E.4    Proof of Theorem 3

Let $f$ be the minimiser of Equation (4). The proof of Theorem 3 separately shows that $f$ has exactly $\left\lceil \frac{n_{k+1}-n_k}{2} \right\rceil \mathbb{1}_{\delta_{n_k-1} \neq \delta_{n_k}}$ kinks on each $(x_{n_k-1}, x_{n_{k+1}})$. Fix in the following $k \geq 0$.

---

[14]Otherwise, the derivative would be weakly decreasing on the interval, contradicting interpolation.

Assume first that $\delta_{n_k-1} = \delta_{n_k}$. Then Lemma 3 along with the definitions of $n_k$ and $n_{k+1}$ directly imply that $s_i^* = \delta_{n_k-1}$ for any $i = n_k, \ldots, n_{k+1} - 1$. This then implies that the associated interpolator, i.e. $f$ has no kink on $(x_{n_k-1}, x_{n_{k+1}})$.

Now assume that $\delta_{n_k-1} \neq \delta_{n_k}$. Without loss of generality, assume $\delta_{n_k-1} < \delta_{n_k}$. By the definition of Equation (10):
$$\delta_j > \delta_{j-1} \quad \text{for any } j \in \{n_k, \ldots, n_{k+1} - 1\}.$$
Moreover, by definition of $n_k$, we have
$$\begin{cases} \text{either } n_k = 1 \\ \text{or } \delta_{n_k-1} \leq \delta_{n_k-2} \end{cases} \quad \text{and} \quad \begin{cases} \text{either } n_{k+1} = n \\ \text{or } \delta_{n_{k+1}} \leq \delta_{n_{k+1}-1} \end{cases}$$

Since $n_{k+1} \leq n_k + 3$ by Assumption 1, Lemma 8 below states that for all the cases, $f$ has exactly $\left\lceil \frac{n_{k+1}-n_k}{2} \right\rceil$ kinks on $(x_{n_k-1}, x_{n_{k+1}})$.

Symmetric arguments hold if $\delta_{n_k-1} > \delta_{n_k}$. In conclusion, $f$ has exactly $\left\lceil \frac{n_{k+1}-n_k}{2} \right\rceil \mathbb{1}_{\delta_{n_k-1} \neq \delta_{n_k}}$ kinks on each $(x_{n_k-1}, x_{n_{k+1}})$. This implies that $f$ has at most
$$\sum_{k \geq 1} \left\lceil \frac{n_{k+1} - n_k}{2} \right\rceil \mathbb{1}_{\delta_{n_k-1} \neq \delta_{n_k}}$$
kinks in total. This concludes the proof of Theorem 3, thanks to Lemma 4.

**Lemma 8.** *For any $k \geq 0$, if $\delta_{n_k-1} < \delta_{n_k}$, then the minimiser of Equation (4) $f$ has*

1. *1 kink on $(x_{n_k-1}, x_{n_{k+1}})$ if $n_{k+1} = n_k + 1$;*

2. *1 kink on $(x_{n_k-1}, x_{n_{k+1}})$ if $n_{k+1} = n_k + 2$;*

3. *2 kinks on $(x_{n_k-1}, x_{n_{k+1}})$ if $n_{k+1} = n_k + 3$.*

Lemma 8 is written in this non-compact way since its proof shows separately (with similar arguments) the three cases.

*Proof.* 1) Consider $n_{k+1} = n_k + 1$. First assume that $x_{n_k} \geq 0$. Lemma 3 implies that $s_{n_k+1}^* \in [\delta_{n_k+1}, \delta_{n_k}]$ and $s_{n_k}^* \in [\delta_{n_k-1}, \delta_{n_k}]$. In particular, $s_{n_k}^* \leq \delta_{n_k}$, which implies that $s_{n_k+1}^* = \delta_{n_k}$. Similarly, $s_{n_k-1}^* \geq \delta_{n_k-1}$, which implies that $s_{n_k}^* = \delta_{n_k-1}$. Using the mapping from Lemma 6, both values $s_{n_k}^*$ and $s_{n_k+1}^*$ yield that the associated function $f$ has exactly one kink on $(x_{n_k-1}, x_{n_k+1})$, which is located at $x_{n_k}$. Similar arguments hold if $x_{n_k} < 0$.

2) Consider now $n_{k+1} = n_k + 2$.

First assume that $x_{n_k+2} < 0$. Thanks to Lemma 3, we can show similarly to the case 1) that $s_{n_k+1}^* = \delta_{n_k+1}$.

Now assume that $x_{n_k+2} \geq 0$. Similarly to the case 1), $s_{n_k+2}^* = \delta_{n_k+1}$. The minimisation problem of the slopes becomes on $s_{i+1}^*$ for $i = n_k$:
$$s_{i+1}^* = \operatorname*{argmin}_{s \in \tilde{S}} g_{i+1}(s, s_i^*) + g_{i+2}(\delta_{i+1}, s),$$
where $\tilde{S} = S_i(s_i^*)$ if $x_{i+1} \geq 0$, and $\tilde{S} = \{\delta_{i+1}\}$ otherwise. Note that $g_{i+1}(s, s_i^*)$ is $\sqrt{1 + x_{i+1}^2}$-Lipschitz in its first argument, while $g_{i+2}(\delta_{i+1}, s) = \sqrt{1 + x_{i+1}^2}|s - \delta_{i+1}|$. Moreover, $s_{n_k}^* \in [\delta_{n_k-1}, \delta_{n_k}]$. As a consequence, either $x_{n_k+1} \geq 0$ and $s_{n_k}^* = \delta_{n_k} = s_{n_k+1}^*$; or $s_{n_k+1}^* = \delta_{n_k+1}$.

Symmetrically, when reasoning on the points $x_{n_k-1}, x_{n_k}$:

- either $s_{n_k}^* = \delta_{n_k-1}$;

- or ($x_{n_k} < 0$ and $s_{n_k}^* = \delta_{n_k} = s_{n_k+1}^*$).

There are thus two possible cases in the end:

- either $(s^*_{n_k} = \delta_{n_k-1}$ and $s^*_{n_k+1} = \delta_{n_k+1})$;

- or $(s^*_{n_k} = \delta_{n_k} = s^*_{n_k+1}$ and $x_{n_k} < 0 \leq x_{n_k+1})$.

In the case where $x_{n_k} < 0 \leq x_{n_k+1}$, we also have $s^*_{n_k-1} = \delta_{n_k-1}$ and $s^*_{n_k+2} = \delta_{n_k+1}$. A straightforward computation then yields a smaller cost on the functions $g_i$ for the choice of slopes $s^*_{n_k} = \delta_{n_k-1}$ and $s^*_{n_k+1} = \delta_{n_k+1}$.

As a consequence, $s^*_{n_k} = \delta_{n_k-1}$ and $s^*_{n_k+1} = \delta_{n_k+1}$ in any case. The mapping of Lemma 6 then yields that $f$ has exactly one kink on $(x_{n_k-1}, x_{n_k+2})$, which is located in $(x_{n_k}, x_{n_k+1})$. Indeed, we either have $a_{n_k-1} = 0$ or $\tau_{n_k-1} = x_{n_k-1}$; similarly either $a_{n_k+1} = 0$ or $\tau_{n_k+1} = x_{n_k+2}$.

3) Consider now $n_{k+1} = n_k + 3$. Similarly to the case 2), we have both

$$\begin{cases} \text{either } s^*_{n_k+2} = \delta_{n_k+2} \\ \text{or } (s^*_{n_k+1} = \delta_{n_k+1} = s^*_{n_k+2} \text{ and } x^*_{n_k+2} \geq 0) \end{cases}$$

$$\text{and} \quad \begin{cases} \text{either } s^*_{n_k} = \delta_{n_k-1} \\ \text{or } (s^*_{n_k} = \delta_{n_k} = s^*_{n_k+1} \text{ and } x_{n_k} < 0). \end{cases}$$

When considering all the possible cases, the mapping of Lemma 6 implies that $f$ has exactly two kinks on $(x_{n_k-1}, x_{n_k+3})$, which are located in $[x_{n_k}, x_{n_k+2}]$. □

## E.5  Adapted analysis for the case $x_1 \geq 0$

This section explains how to adapt the analysis of this section to the easier case where all $x$ are positive. Lemma 7 holds under the exact same terms (but its second part is useless) in that case. From there, the proof of Theorem 2 consists in just showing the uniqueness of the minimisation problems for any $\tilde{\mathbf{s}} \in \mathcal{S}$:

$$\min_{s_{i_0} \in \mathbb{R}} c_{i_0}(s_{i_0})$$

$$\min_{s_{i+1} \in \mathcal{S}_i(\tilde{s}_i)} g_{i+1}(s_{i+1}, \tilde{s}_i) + c_{i+1}(s_{i+1}) \quad \text{for any } i \in \{i_0, \dots, n-1\}.$$

The unique solution of the first problem is $\delta_{i_0}$ thanks to Lemma 7, while same arguments as in Appendix E.1 hold for the second problem.

For the proof of Lemma 3, the exact same arguments as in Appendix E.2 hold for any $i \geq i_0 + 1$. For $i = i_0 = 1$, it is obvious in that case that $s^*_1 = \delta_1$, leading to Lemma 3.

Finally, the proof of Theorem 3 follows the same lines when $x_1 \geq 0$.

## E.6  Proof of Corollary 1

*Proof of Corollary 1.* For classification, the natural partition to define is the following:

$$n_1 = 1 \text{ and for any } k \geq 0 \text{ such that } n_k < n+1,$$

$$n_{k+1} = \min \{j \in \{n_k + 1, \dots, n\} \mid y_{n_k} \neq y_j\} \cup \{n+1\}. \tag{36}$$

This partition splits the data so that for any $k$, $y_i$ has a the same value for $i \in \{n_k, n_{k+1} - 1\}$. Denote $K$ the number of $n_k$ defined in Equation (36), i.e., $n_K = n + 1$. From there, by simply noting that any margin classifier has at least a kink in $[x_{n_k}, x_{n_{k+1}})$ for $k \in [K-2]$:

$$\min_{\substack{f \\ \forall i \in [n], y_i f(x_i) \geq 1}} \|f''\|_0 = K - 2.$$

Similarly to the proof of Lemma 1, we can first show the existence of a minimum.[15] Let us now consider $f$ a minimiser of

$$\min_{\substack{f \\ \forall i \in [n], y_i f(x_i) \geq 1}} \left\| \sqrt{1+x^2} f'' \right\|_{\mathrm{TV}}. \tag{37}$$

Define the set

$$S = \left\{ n_k \mid k \in \{2, \ldots, K-1\} \right\} \cup \left\{ n_k - 1 \mid k \in \{2, \ldots, K-1\} \right\}.$$

By continuity of $f$, we can choose an alternative training set $(\tilde{x}_i, \tilde{y}_i)$ satisfying:

$$\tilde{x}_i \in [x_{n_k-1}, x_{n_k}] \quad \text{for any } i \in \{n_k - 1, n_k\},$$
$$y_i = f(\tilde{x}_i) \quad \text{for any } i \in S.$$

Then, a direct application of Theorem 2 yields that the minimisation problem

$$\min_{\substack{\tilde{f} \\ \forall i \in S, y_i = \tilde{f}(\tilde{x}_i)}} \left\| \sqrt{1+x^2}\, \tilde{f}'' \right\|_{\mathrm{TV}}, \tag{38}$$

admits a unique minimiser, that we denote $f_{\mathrm{reg}}$. But also note that this unique minimiser is also in the constraint set of Equation (37) thanks to Lemma 3, so that

$$\left\| \sqrt{1+x^2} f''_{\mathrm{reg}} \right\|_{\mathrm{TV}} \geq \left\| \sqrt{1+x^2} f'' \right\|_{\mathrm{TV}}.$$

However, since $f$ is in the constraint set of Equation (38), we actually have an equality, and by unicity of the minimiser of Equation (38),

$$f_{\mathrm{reg}} = f.$$

Moreover, it is easy to check that Assumption 1 holds for the data $(\tilde{x}_i, y_i)_{i \in S}$, with $n_{k+1} = n_k + 2$. As a consequence, Theorem 3 implies that the minimiser of Equation (38) is among the sparsest interpolators for the set $(\tilde{x}_i, y_i)_{i \in S}$, i.e. it exactly counts $K - 2$ kinks. This then implies that $\|f''\|_0 = K - 2$, so that

$$\underset{\substack{f \\ \forall i \in [n], y_i f(x_i) \geq 1}}{\mathrm{argmin}} \ \bar{R}_1(f) \subset \underset{\substack{f \\ \forall i \in [n], y_i f(x_i) \geq 1}}{\mathrm{argmin}} \ \|f''\|_0. \tag{39}$$

$\square$

---

[15] Proving that the sequence $(a_j, b_j)$ is bounded is here a bit more tricky. Either the data is linearly separable, in which case the minimum is 0, or the data is not linearly separable. When the data is not linearly separable, then $(a_j, b_j)$ is necessarily bounded, since $(\mu_j, a_j, b_j)$ would behave as a linear classifier for arbitrarily large $(a_j, b_j)$.

