# OpenReview forum: "Penalising the biases in norm regularisation enforces sparsity"
_NeurIPS.cc/2023/Conference — NeurIPS 2023 poster_

### Official Review · Reviewer_3DUe · 2023-07-05

**Soundness:** 3 good
**Presentation:** 3 good
**Contribution:** 2 fair
**Rating:** 6
**Confidence:** 3

**Summary:**

This work considers a single hidden layer ReLU network $f_\theta: \mathbb{R} \rightarrow \mathbb{R}$ with biases in the hidden layer as well as skip connections. In this context the representational cost $R(f)$ required to interpolate a function $f: \mathbb{R} \rightarrow \mathbb{R}$ is studied, which is defined as the infinimum of the 2-norm of the network parameters such that $f_\theta = f$. Following prior work and in order to consider the larger space of functions that can be approximated to an arbitrary degree by such finite width networks, (potentially) infinite width networks can be parameterized by a measure on the unit sphere, allowing the representation cost to be recast as the total variation of this measure such that $f_\mu = f$. Unlike prior work, the the bias terms of the hidden layer are included in the norm penalty. As a result of this, for a univariate Lipschitz function $f$ the representation cost is the total variation of $\sqrt{1-x^2}f''$ instead of just $f''$ when the bias terms are not included. Given a training sample $(x_i, y_i)_{i=1}^n$ the minimum-norm (with respect to the the total variation of $\sqrt{1-x^2}f''$) of interpolators $f$ such that $f(x_i) = y_i$ are studied. In Theorem 2 it is shown that the minimum-norm interpolator in this context is unique, which is not necessarily the case when the bias terms are not included. Furthermore, assuming the data is such that the linear interpolation between any consecutive set of 6 points is neither convex or concave, then in Theorem 3 it is shown that the minimum norm interpolator has a minimal number of kinks.

**Strengths:**

This work builds on prior work by studying the impact of including the bias parameters in the 2-norm parameter regularization term. In particular, the min-norm interpolator in this context is unique and the results also suggest including the bias terms perhaps encourages the network towards finding sparser network solutions in terms of the number of kinks or bumps. The work appears technically sound however I did not thoroughly read the appendices. The paper is also well written and structured. The contribution appears tangible if somewhat incremental.

**Weaknesses:**

1. One of the key findings over prior works is that regularizing the bias terms encourages a simpler interpolating function with only a few kinks. However, this relationship is only proved in a restrictive data setting where the linear interpolation between a small number of consecutive points is neither convex or concave. This feels like quite a restriction, furthermore, from the discussion lines 427-432 of the supplementary material, it seems at least experimentally that when the data does contain convex or concave regions then the minimum norm interpolator and sparsest interpolators are not really related. Therefore I wonder whether this takeaway is really a general principle or more a quirk only exhibited in a highly specific setting.

2. The setting, i.e., univariate data, the necessity for skip connections whose parameters are not included in the penalty is also restrictive, albeit  in line with other works. The focus on studying the min-norm interpolators also does not capture learning.

3. Another minor comment I would have is that the statement to the effect of ``the most important factor behind generalization is the norm of the parameters" in the introduction is a bit strong in my opinion. I would argue the jury is still out on this, in practice one can fairly often train a network with no explicit regularization and it can do fine and there is also a line of work suggesting implicit regularization cannot be explained in terms of norm minimization, see e.g., https://arxiv.org/pdf/2005.06398.pdf, https://arxiv.org/abs/2012.05156.pdf.

**Questions:**

1. Does one observe learning a smoother function, i.e., one with less kinks or bumps experimentally when the bias term is included versus not included and how does this vary with say the regularization parameter as well as the smoothness of the target function?

2. If one considers deeper networks or multivariate inputs or a different loss function etc. is there still evidence of a tendency towards learning a function with fewer bumps when bias terms are included in the penalty?

3. Empirically simulating across a number of trials what is the distribution of the learned function look like? Does it concentrate around the minimum norm interpolator?

3. In general how sensitive are the findings of this work to say to changes in network architecture (e.g., different activation function), moving to higher dimensional inputs, changing the loss function from squared error to cross-entropy say?


**Limitations:**

A brief discussion of limitations is provided in the conclusion avenues for future work are identified. I cannot foresee any negative societal outcomes from this work.

---

> ### Author Rebuttal · Authors · 2023-08-07
>
> We thank the reviewer for the insightful comments. We now answer the concerns.
>
> *Incremental contribution*
> ===
> In the theory of NNs, the considered settings are often incremental wrt previous works. This incremental approach stems from the difficulty of the considered problems. Even relaxing a small condition can pose significant technical challenges.
>   The proof of Theorem 3, in particular, requires entirely new analyses and proof techniques. Thus, while the setting may be incremental, we firmly believe that Theorem 3 and the employed techniques, notably the dynamic program equivalence, extend well beyond an incremental scope.
>
> *Assumption 1*
> ===
> Please refer to our response to Rev. ZvaF for more details. The fact that min norm interpolators are not always the sparsest ones surprised us, as we had long believed that Theorem 3 would hold without any data condition. It was only after multiple experiments that we realized the min-norm interpolator may not be the sparsest. As a consequence, we still consider Theorem 3 to be significant. It sheds light on unexpected behaviors, which were not initially apparent.
> We believe that this discrepancy between min norm interpolators and sparsest interpolators also holds significance in gaining a better understanding of the implicit bias phenomenon (see our response to Rev. ZvaF and below). Additionally, it is worth noting that the result remains applicable for binary classification (Coro 1).
>
>   Whether this takeaway is a general principle remains an open question. We here only address this question within a simple and tractable setting.
>
> *Restrictive setting*
> ===
> As highlighted by Rev. fxbg, relaxing any of these conditions proves to be considerably more challenging. As often when studying NNs, comprehending the intricate and general setups used in practice remains elusive. Therefore, we focus on simpler settings that allow for precise quantification of already non-trivial phenomena. Please note that the considered simplifications (shallow networks, skip connections, univariate setups) are common in the literature.
>
> As you pointed out, this work does not focus on optimization but rather on describing the min-norm interpolator. Both aspects are complementary and can be treated independently. Min-norm interpolators seem to play a pivotal role, given their close relation to various implicit bias works (on the optim aspect).
>
> *Parameters norm for generalization*
> ===
> In the NN literature, most generalization bounds rely on the parameters norm. The networks' ability to generalize, even without explicit regularization, is largely attributed to the implicit bias inherent to the optim algo. Although the implicit bias may not always favor the min-norm interpolator (see below), it generally guides the networks toward small-norm solutions (if not min-norm).
>
> *Implicit bias $\neq$ min norm*
> ===
> We agree that implicit bias might not always lead to min-norm interpolators, although it appears to have a close association.
>
> Besides the mentioned references, a recent paper (Chistikov et al. 2023) presents a multidim example of data where the min-norm interpolator does not align with the sparsest interpolator. In this example, they demonstrate that the implicit bias instead leads to a sparsest interpolator (please see our answer to Rev. ZvaF for more details).
>
> Characterizing the exact criterion minimized by the implicit bias is an open research direction. Perhaps the implicit bias is better captured by “sparsest interpolator” and situations where “sparsest $\neq$ min norm” correspond to cases where implicit bias does not lead to min-norm interpolators. Overall, the relationships between sparsest, min-norm interpolators and implicit bias remain misunderstood. We believe that our work contributes to advancing their understanding.
>
> *Smoothnes*
> ===
> As depicted in Figure 3, a smoother function is indeed learned when the biases are unpenalized.  This does not vary much with the regularization parameter. In fact, for larger values of this parameter, we may even observe fewer kinks, as the obtained estimator then does not overfit.  Unlike kernel regression settings, there is no target function here; only a finite dataset is considered.
>
> *Empirical evidence in harder cases*
> ===
> Conducting an empirical study on real data and architectures demands substantial effort and resources. Nevertheless, we believe the success of distillation is likely connected. For instance, the experiments conducted by Hinton et al. (2015) highlight that the learnt estimator  can be replicated by a smaller network. This aligns with our results, as the number of kinks is the number of neurons necessary to represent the function.
>
> *Sensitivity to the setting*
> ===
> In theory, the activation and single hidden layer play a critical role in obtaining the infinite width representation, as discussed with Rev. fxbg.  Analyzing higher dimensional inputs becomes considerably more intricate. An equivalent version of Theorem 1 has been proposed by Ongie et al. (2019) for the multidimensional case when biases are omitted. It might be possible to extend such a result when counting the biases, but seems much harder.
>
>  On the other hand, deriving a multidimensional version of Theorem 3 remains even unknown when bias terms are omitted.
>
> *Other losses*
> ===
> For cross-entropy loss, Corollary 1 answers the question and shows that no data assumption is required for classification. These results are not influenced by the loss function, but rather by the setting (regression vs classif). In regression, the estimator of interest is the min-norm interpolator, applicable to most regression losses. Conversely for classif, the estimator of interest is the min-norm margin classifier, relevant for most classification losses.
>
> *Concentration around min norm*
> ===
> If we examine the distribution of the function learnt in Figure 3b) over multiple rounds, it won't concentrate around the function in Figure 3a).

---

> > ### Comment · Reviewer_3DUe · 2023-08-14
> >
> > Thanks for answering my questions, on balance I will keep my current score.

---

### Official Review · Reviewer_brcW · 2023-07-07

**Soundness:** 4 excellent
**Presentation:** 4 excellent
**Contribution:** 3 good
**Rating:** 7
**Confidence:** 4

**Summary:**

This paper studies the representational cost of a toy ReLU MLP with a single hidden layer, and a scalar input. The main result of the paper is that, when regularizing both the weights and the biases, the minimum cost functions are min norm interpolators which are also sparse (defined in the paper as having the minimal number of kinks). When the bias is not regularized (as in the standard weight decay), the minimum cost functions are not necessarily sparse.

**Strengths:**

Overall i enjoyed reading this paper. Its main strengths are:
1. Very well written, and positioned relative to prior work.
2. A rigorous, clean result, albeit on a very toyish model
2. A novel result that is somewhat surprising and non-trivial.

**Weaknesses:**

The main drawbacks of the paper are:
1. The studied model is very simple, even for a theory paper (1D input)
2. Its not clear whether the main conclusion of the paper has any bearing in practice. Therefore, its impact might be restricted to the theoretically inclined.

These drawbacks are somewhat nitpicky though, and i do not see a reason not to accept it.

**Questions:**

Although the sparsest interpolator is not the only solution when the bias is not regularized, we do see (figure 3b) that standard weight decay does produce sparse (almost sparsest) interpolators. Is it possible that in practice the effect of regularizing the bias is not that pronounced?

**Limitations:**

Limitations are adequately addressed

---

> ### Author Rebuttal · Authors · 2023-08-07
>
> We thank the reviewer for the insightful comments. We now answer the raised concerns.
>
>
>   We agree with the weaknesses pointed out by the reviewer. As discussed in the paper, we indeed consider a simplified setting (1d, skip connection, etc.) as it allows for a tractable analysis. Studying a more general setting would be much harder, as pointed out by Reviewer fxbg and we believe that we lack the necessary  understanding and tools at the moment.  We also agree that the main interest of this work is theoretical. In particular, we believe it contributes to the understanding of implicit bias, min norm interpolators and their generalization properties. Actually, our work has already proven useful to show some catastrophic overfitting behavior of min norm interpolator in the presence of noise in the very recent paper [1].
>
> *Practical effect of regularizing the bias*
> ===
> The empirical effect of regularizing the bias indeed seems subtle in practice. We believe that this subtlety arises due to the influence of the implicit bias of the optimization algorithm which still regularizes the bias weights. Indeed, given the nature of the optimization updates, the implicit bias treats the bias weights exactly as the other weights. It is precisely because the implicit bias accounts for the bias term that we believe the true interpolator of interest does count the biases.  The primary objective of this work is thus to gain a deeper understanding and provide characterization of this particular interpolator, primarily from a theoretical standpoint.
>
> ——————————————
>
>  References.
> [1] Joshi, N., Vardi, G., & Srebro, N. (2023). Noisy Interpolation Learning with Shallow Univariate ReLU Networks.

---

> > ### Comment · Reviewer_brcW · 2023-08-18
> >
> > Thank you for your response. I will keep my positive score as i think the results of this paper are worth publishing

---

### Official Review · Reviewer_fxbg · 2023-07-20

**Soundness:** 3 good
**Presentation:** 3 good
**Contribution:** 2 fair
**Rating:** 5
**Confidence:** 3

**Summary:**

The paper addresses two main questions: (1) Can the parameters' norm be understood in terms of a quantity that is more insightful from a functional analysis point of view? (2) What kind of functions are learned when minimizing the empirical loss, either with an explicit or implicit regularization of the parameters' norm? The paper provides an answer to these questions for univariate functions when the bias parameters are regularized. It shows that penalizing the bias terms enforces sparsity and uniqueness of the estimated function, which does not hold without penalization of the biases. The paper provides a theorem (see Theorem 1) that states the representational cost of a function is given by the total variation of its second derivative, weighted by an sqrt(1 + x^2) factor. This weighting does favor sparse estimators when training neural networks.

**Strengths:**

- It is well-known that omitting the bias parameters of the NN makes the analysis easier. So, the fact that the bias terms are included is highly welcomed. There are interesting conclusions garnished from including the bias terms, such as the enforcement of sparsity.

- The paper shows that the representational cost of a function is given by the total variation of its second derivative, weighted by an sqrt(1 + x^2) factor.



**Weaknesses:**

- The work is only for univariate functions, mainly for ReLU NNs with skip connections, penalization terms (that are not common in practice), and continuous dynamics for training. Therefore, the theoretical study here may say very little about the answers to the manuscript's two main questions for practical NNs.  However, I acknowledge that it may be extremely hard to do analysis when the setup is modified and the authors provide empirical evidence.

- A skip connection is a useful architecture to make the analysis easier; however, whether it says anything about the NNs that are used in practice is unclear to me. The author says: "Since a skip connection can be represented by two ReLU neurons, it is commonly believed that considering a free skip connection does not alter the nature of the obtained results."; however, I am not aware of a theoretical statement that makes it precise in this setting.


**Questions:**

1. I wonder if there is anything special to the weight sqrt(1 + x^2) from a higher level and how particular this result is to the setup considered in the paper, e.g., Relu, Skip connection, 1D setup, and one hidden neural network.

2. The definition of the parameter norm is the squared $\ell_2$ norm of all the {a_j}, {b_j}, and {w_j}. While in 1D, we can consider them as 3 different vectors, but in higher dimensions, I am not sure about viewing them as vectors is the correct way of defining the parameter norm. Also, when it comes to sparsity, one would compute the number of its nonzero elements, an alternative of which is the \ell_1 norm. Using the squared $\ell_2$ norm as a sparsity measure is an uncommon choice since a neural network can be non-sparse at all but with a very small parameter squared $\ell_2$ norm.

3. It is not immediately clear from the paper what type of function f being considered in the main result. It was said that the second derivative of f, i.e., f'' should be understood in terms of distribution, which I assume that non-smooth 1D functions are considered. Based on the 2nd definition of \bar{R}_1(f), f is restricted to those represented by a simple Relu NN.

4. It is written that the two \bar{R}_1(f) in Section 2.1 can be proven to be the same. Can the authors provide proof or pointing some references to it?

**Limitations:**

There are some comments on the future directions of this work in the conclusion, but I am not sure they are classified as limitations of this work.

Here is the main limitation in my perspective of the paper:

1. The analysis is restricted to a 1D setting with a very particular structure for the neural network. The weight sqrt(1 + x^2) is too particular, and I am not confident that one can return the same result with a slight change in the problem setup.

2. The implication of minimizing the TV norm of sqrt(1 + x^2) f''(x) is not well understood, and its link to sparsity promoting is not so clear.

---

> ### Author Rebuttal · Authors · 2023-08-07
>
> We thank the reviewer for the many insightful comments. We now answer the raised concerns.
>
> *Restricting setting*
> ===
> As you rightly highlighted, analyzing any of these potential extensions is undoubtedly much more challenging. When conducting theoretical studies on neural networks, it is often difficult to fully grasp the complexities and general configurations encountered in practical applications. Consequently, we tend to concentrate on simpler settings where we can accurately describe and quantify already non-trivial phenomena.
>
> Additionally, it is worth noting that the settings we have considered, such as shallow networks, skip connections and univariate setups are commonly encountered in the literature. These simplifications serve as valuable starting points for delving into the underlying principles and behaviors of neural networks.
>
> *Skip connection*
> ===
> We emphasize that skip connections are used in numerous state-of-the-art architectures. They were first popularized in ResNet models, which have maintained their competitiveness over time. Presently, many of the top-performing models are based on transformers, and these architectures also incorporate skip connections.
>
> Furthermore, a skip connection corresponds to a linear function $x\mapsto ax +b$. Its equivalence with two ReLUs can be directly derived using the following identity: $ ax+b = \sigma(ax+b) - \sigma(-ax-b).$
>
> As a consequence, we believe that adding/removing a free skip connection does not impact much the sparsity (in the number of kinks) of the network, as a skip connection can be represented by only two neurons. This observation is confirmed by the experiments of Figure 3, as they were done without free skip connection.
>
> *Specific to the setup*
> ===
> As most of the literature, the results are indeed specific to the ReLU activation and single hidden layer architecture. The positive homogeneity property of ReLU activation is essential in deriving the equation provided after line 101. A single hidden layer on the other hand allows to give a nice separation in the neurons interactions (i.e; the output weight $a_j$ only interacts with the weights $w_j,b_j$).
>
> When either the homogeneity property is absent or the network has more layers, the correspondence between finite width networks and infinite width representations remains unknown.
>
> On the other hand, there is hope that our results (or similar ones) may hold  beyond the 1D setup and skip connection. Nonetheless, the extensions proves to be challenging, as you already pointed out.
>
> *About the $\sqrt{1+x^2}$ weight*
> ===
> It actually corresponds to the norm of the vector $(x,1)$. In the implicit bias literature, a common parameterization involves using lifted data $\tilde{x}=(x,1)$ to account for the bias weights. This term thus accounts for the norm of the lifted data point. The origin of the $\sqrt(1+x^2)$ term may appear puzzling, but is better understood when considering the representational cost of a single (or multiple) neuron(s) network.
>
> Consider the network $f(x)=a\sigma(wx+b)$. A simple computation yields $f’’=aw \delta_{\frac{-b}{w}}$, were $\delta_{x}$ is Dirac distribution located at $x$. Computing $||\sqrt{1+x^2}f’'||$ then yields
> $||\sqrt{1+x^2}f’’||=|a|\sqrt{w^2+b^2},$  which also turns out to be the minimal representation cost of $f$ (e.g. using equation after line 101).
>
> *The $\sqrt{1+x^2}$ weight is too particular*
> ===
> We note this weight is not chosen on purpose, but results from the setting. As explained above, the problem setup is indeed restricted to enable a manageable and tractable analysis. At present, we do not have a clear approach for tackling more challenging settings.
>
> *Sparsity measure*
> ===
> The $\ell_2$ norm of the parameters is closely linked to sparsity in neural networks, owing to the $2$-homogeneity of the parameterization. This homogeneity allows the reformulation as an $\ell_1$ norm of the output layer (see equation after line 101), which then draws parallels to the conventional Lasso estimator in linear regression.
>
>  For a more comprehensive understanding of our concept of functional sparsity, defined as the number of kinks of the represented function, we recommend referring to our response to Reviewer ZvaF, where we provide further details.
>
> Finally, in cases involving multiple dimensions, the literature typically views the weights $w$ as a $d\times m$ matrix, extending the analysis to higher-dimensional scenarios.
>
> *Considered function space*
> ===
> The paper considers *any* type of univariate function $f:\mathbb{R}\to\mathbb{R}$. More precisely, Theorem 1 specifies that the equality only holds for Lipschitz functions (otherwise $\bar{R}_1(f)$ is infinite).  As explained in the introduction, the second definition of $\bar{R}_1(f)$ is for functions that can be represented as *infinite width* neural networks ($\mu$ is a signed measure). This set is much larger than functions represented by a simple (finite) ReLU neural network. However, this definition is even more general and remains valid for any function: when a function cannot be represented by an infinite neural network, the infimum is taken over the empty set, resulting in $\bar{R}_1(f)=+\infty$.
>
> *Equality of the two $\bar{R}_1(f)$*
> ===
> All results presented in Section 2, including this one, follow the same lines as the preliminary results from Savarese et al. (2019), Ongie et al. (2019). There are only minimal adjustments required to account for the norm of the bias terms, with no additional complexity introduced.
>
> Our work precisely aims to improve the understanding of the link between minimizing the TV norm and sparsity promoting, which is currently not well-established. In particular, we derive conditions under which the min TV norm interpolator corresponds to a sparsest interpolator. We also show that without such conditions, the min TV norm one might not be among the sparsest interpolators.

---

> > ### Comment · Reviewer_fxbg · 2023-08-18
> >
> > I want to thank the authors for answering my questions/comments. I decided to raise my score.

---

### Official Review · Reviewer_ZvaF · 2023-07-26

**Soundness:** 4 excellent
**Presentation:** 3 good
**Contribution:** 2 fair
**Rating:** 5
**Confidence:** 3

**Summary:**

This paper theoretically investigates the effect of regularizing bias parameters’ norm, mostly ignored or omitted in previous analyses, for univariate one hidden layer ReLU networks. It turns out that minimizing $L_2$ norms for both weights and biases corresponds to regularizing a functional norm defined as the total variation of the second derivative of the function modeled by the neural network with a weight of $\sqrt{1+x^2}$ factor, where the weight factor does not exist when regularizing only the weight norm. The minimal functional norm interpolator can be obtained from a proposed dynamic programming algorithm, and the uniqueness and sparsity properties of the interpolator are discussed.

**Post rebuttal** I have increased my score to 5 from 4. I can appreciate the theoretical contribution, while the practical relevance still seems to be a weakness.

**Strengths:**

- The paper analyzes the effect of bias norm regularization, which has not been considered much before, showing novel and original results. The effect is not trivial, and the regularization problem is identified as minimizing a functional norm with a weight factor, which leads to interesting uniqueness and sparsity properties of minimum norm interpolators.
- Proofs are provided for most of the theorems and lemmas in a mathematically sound way. It is especially ingenious to prove the uniqueness and sparsity properties of the minimum norm interpolator by using a dynamic programming algorithm that obtains the interpolator.


**Weaknesses:**

- The benefit of the considered sparsity regarding the number of a function’s kinks is unclear. Why would the number of kinks be important? Can it be used to explain generalization properties in any way? Also, the sparsity in terms of kinks seems quite different from the one considered in usual machine learning contexts, e.g., having zero values for most weights or representations except for a few. Maybe addressing these points and providing some intuitive explanations would be helpful to enforce the claim of the paper.
- The condition for the minimum norm interpolator to be the sparsest interpolator is quite restrictive according to **Assumption 1**. Moreover, **Appendix A** provides an example of when the sparsest and minimal norm ones are largely different in the number of kinks. These points can lower the significance of the sparsity result.
- Due to the weight factor of $\sqrt{1+x^2}$ in evaluating $\mathcal{F}_1$ norm from bias regularization, the shift invariance property of the norm disappears. Can this have any negative impact on using the function norm to explain the nature of a function? Also, the connection between this weight factor and the sparsity property regarding the number of kinks is not intuitive. Rather, it seems that containing this weight factor would make the location of kinks not far from zero.


**Questions:**

In addition to the questions posed in the **Weaknesses** section, I have the following questions:
- What would be the practical benefit of considering bias regularization? As far as I know, a common practice for applying weight decay in neural network training is only regularizing the weight norm. This method has already shown many empirical successes so far.

**[Minor comments]**
- How is the ‘activation cone’ defined in line 188?
- Providing some explanations about deriving (5) would be helpful.
- **Corollary 1** is corrected in **Appendix E.6**, and I think the corrected version should come to the main text.


**Limitations:**

The limitations of the results are discussed in **Conclusion** and **Appendix A**. For other possible limitations, please refer to the weaknesses and questions above.

---

> ### Author Rebuttal · Authors · 2023-08-07
>
> We thank the reviewer for the many insightful comments. We now answer the raised concerns.
>
> *Sparsity*
> ===
> The generalization benefit of the considered sparsity remains unclear in the literature. Indeed, generalization bounds rely on the parameters’ norm rather than the network width (ie sparsity level); and understanding the generalization properties of sparsest neural networks remains an ongoing research direction.
>
> However the primary objective of our work is not about the generalization of sparse networks but instead to fully characterize the min norm interpolators. These min norm interpolators are of crucial interest and happen to be linked with sparsity.
>
> This notion of sparsity is natural, as it corresponds to the minimal width necessary for a network to represent the function. Counting the number of non-zero weights is less relevant in that context, since networks with different weights can represent the same function. For instance in Figure 3a), a network with hundreds of non-zero weights is equivalent to a width 2 network.
>
> Specifically, our notion of sparsity counts the minimal number of non-zero triplets $(a_j,b_j,w_j)$ required to represent the function. It thus quantifies the best possible level of sparsity *of the weights*. We emphasize that evaluating quantities based on the represented function, rather than the network weights, is more relevant since it is the genuine object of interest.
>
>  *Assumption 1*
> ===
> Finding that min norm interpolators are not always the sparsest ones came as a surprise to us. We previously held the belief that Theorem 3 would hold without any data condition. It was only after multiple experiments that we discovered the min-norm interpolator's inability to always be the sparsest. We still consider Theorem 3 to be of utmost importance, since it sheds light on unexpected behaviors, which were not initially apparent.
>
>   Furthermore, this discrepancy between min norm and sparsest is crucial in (Chistikov et al. 2023). The authors present a multidimensional example where the min norm interpolator is not the sparsest. On this example, the implicit bias of gradient flow does not favor the min norm interpolator, but instead leads to the sparsest one, contradicting some previously held beliefs. These findings are closely related to the works referenced by Rev. 3DUe, which also argue that the implicit bias does not consistently favor min norm interpolators. The ongoing exploration of these phenomena constitutes an intriguing and open research direction: perhaps the implicit bias is better captured by “sparsest interpolators” and instances where “sparsest $\neq$ min norm” correspond to scenarios where the implicit bias does not lead to min norm interpolators.
>
>  In summary, the intricate relationships between sparsest, min-norm interpolators and implicit bias are far from being fully comprehended. We believe that our work contributes to advancing their understanding.
>
> Note however that for (binary) classification, Corollary 1 *always*  yields sparsest margin = min norm margin, which also aligns with the implicit bias.
>
> *Shift invariance*
> ===
> When counting the bias terms, the considered norm is indeed no longer shift invariant, which complexifies the analysis techniques. However, preserving strict shift invariance may not be crucial in practical applications, since centering the data is a common practice. Interestingly, Nacson et al. (2022) show that dealing with uncentered data might be beneficial for learning relevant features. The lack of shift invariance in this context can lead to improved performance, thanks to better feature learning.
>
> *Intuition on $\sqrt{1+x^2}$ weighting*
> ===
> The initial intuition might indeed suggest that having kinks as close as possible to zero would be the best strategy. However, upon closer examination, such an approach results in larger amplitudes for subsequent kinks.
>
>  Finding the right balance is not as straightforward: kinks near zero are favored by the direct cost component, while it also affects the amplitude of the next kink(s). This balance is highlighted by the dynamic program: $g$ takes into account the direct cost, encouraging kinks near zero, while $c$ reflects the influence of choosing the current kink on the next one, favoring smaller amplitudes in subsequent kinks.
>
> *Practical regularization of bias*
> ===
> There is limited investigation in the literature on the distinction between counting and non-counting bias terms. Our study is mainly theoretical and does not assert the superiority of any specific rule in practice. We believe that in practice, both schemes yield similar results, because of the presence of additional implicit regularization.  Omitting the bias terms might then be preferred as it reduces computation time. Also, omitting biases might be preferred for optimization reasons (see footnote 1), which is not the focus of our work.
>
> The influence of our work is mostly theoretical, as substantial theoretical distinctions may arise between the considered interpolators. This difference is due to a large gap between the worst-case min norm interpolator (omitting bias) and the unique min norm interpolator (counting bias).
>
> Nevertheless in practical applications and owing to the implicit bias, the estimator obtained omitting bias does not represent the worst-case scenario but instead approximates the unique min norm interpolator (counting bias).
>
>    *Minor comments:*
> ===
> * In 1D, the activation cones are defined line 521
> * The idea to derive Eq (5) is by first using Lemma 1. From then, the minimal cost for connecting $(x_{i+1}, y_{i+1})$ with $(x_i, y_i)$ is done by using a single kink in between. The conditions on $s_i$ and $s_{i+1}$ then yield a unique possible kink. Minimizing its neuron norm then yields Eq (5)
> * We spotted an error in Corollary 1 after the main deadline and were only able to correct it in the supplementary material. It will of course be corrected in the revised version.

---

> > ### Comment · Reviewer_ZvaF · 2023-08-18
> >
> > I appreciate the authors' responses. I will increase my score.

---

### Decision · Program_Chairs · 2023-09-21

**Decision:**

Accept (poster)

**Comment:**

This paper continues the study of minimum norm univariate networks, now showing a sparsity sensitivity. Reviewers are overall positive but have many comments, I urge the authors to revise appropriately during the camera ready phase.